# The vertical distribution of biomass burning pollution over tropical South America from aircraft in situ measurements during SAMBBA

Eoghan Darbyshire[1], William T. Morgan[1], James D. Allan[1], Dantong Liu[1], Michael J. Flynn[1], James R. Dorsey[1], Sebastian J. O'Shea[1], Douglas Lowe[1], Kate Szpek[2], Franco Marenco[2], Ben. T. Johnson[2], Stephane Bauguitte[3], Jim M. Haywood[2,4], Joel F. Brito[5*], Paulo Artaxo[5], Karla M. Longo[6**], Hugh Coe[1]

[1]Centre for Atmospheric Science, University of Manchester, Manchester, UK
[2]Met Office, Exeter, UK
[3]Facility for Airborne Atmospheric Measurements (FAAM), The University of Cranfield, Cranfield, UK
[4]CEMPS, University of Exeter, Exeter, UK
[5]Physics Institute, University of Sao Paulo, Sao Paulo, Brazil
[6]National Institute for Space Research (INPE), Sao Jose dos Campos, Brazil
*Now at Laboratory for Meteorological Physics (LaMP), University Clermont Auvergne, Aubière, France
**Now at NASA Goddard Space Flight Center and USRA/GESTAR, Greenbelt, MD, USA

*Correspondence to:* Eoghan Darbyshire (eoghan.darbyshire@manchester.ac.uk)

**Abstract.**

We examine processes driving the vertical distribution of biomass burning pollution following an integrated analysis of over 200 pollutant and meteorological profiles measured in-situ during the South American Biomass Burning Analysis (SAMBBA) field experiment. This study will aid future work examining the impact of biomass burning on weather, climate and air quality.

During the dry season there were significant contrasts in the composition and vertical distribution of haze between western and eastern regions of tropical South America. Owing to an active or residual convective mixing layer, the aerosol abundance was similar from the surface to ~1.5 km in the west and ~3 km in the east. Black carbon mass loadings were double in the east ($1.7$ µg m$^{-3}$) than west ($0.85$ µg m$^{-3}$) but aerosol scattering coefficients at 550 nm were similar (~120 Mm$^{-1}$), as too were CO near surface concentrations (310-340 ppb). We attribute these contrasts to the more flaming combustion of Cerrado fires in the east and more smouldering combustion of deforestation and pasture fires in the west. Horizontal wind shear was important in inhibiting mixed layer growth and plume rise, in addition to advecting pollutants from the Cerrado regions into the remote tropical forest of central Amazonia. Thin layers above the mixing layer indicates roles for both plume injection and shallow moist convection in delivering pollution to the lower free troposphere. However, detrainment of large smoke plumes into the upper free troposphere was very infrequently observed. Our results reiterate that thermodynamics control the pollutant vertical distribution and thus point to the need for correct model representation so the spatial distribution and vertical structure of biomass burning smoke is captured.

We observed an increase of aerosol abundance relative to CO with altitude both in the background haze and plume enhancement ratios. It is unlikely associated with thermodynamic partitioning, aerosol deposition or local non-fire sources. We speculate it may be linked to long range transport from West Africa or fire combustion efficiency coupled to plume injection height. Further enquiry is required to explain the phenomenon and explore impacts on regional climate and air quality.

# 1 Introduction

The vertical distribution of biomass burning pollution influences its impacts on weather (e.g. Kolusu et al., 2015), climate (e.g. Boucher et al., 2013), human health (e.g. Reddington et al., 2015) and ecosystem function (e.g. Rap et al., 2015). The vertical location of aerosol is particularly sensitive since this can impact radiative forcing (Zarzycki and Bond, 2010), alter cloud microphysics (Tao et al., 2012) and feedbacks on cloud dynamics (Feingold et al., 2005). The altitude of pollutants also determines their atmospheric residence time, which will affect any ageing processes and the resultant horizontal distribution following advection. Model predictions of the vertical distribution of the effects of biomass burning smoke are therefore very dependent on accurately capturing the vertical structure, especially in regions of the tropics where land use change is rapid and smoke is widely distributed. However, a lack of detailed observations across these regions limits model constraint.

These uncertainties are pronounced in tropical South America (TSA), one of the largest global biomass burning sources. Aerosol accumulates within the convective boundary layer forming a regional haze that can cover up to 6 million km$^2$ (Prins et al., 1998) and whilst weekly averaged aerosol optical depths (AOD) are typically 0.75-1 in the mid visible, they can reach 4 in extremely polluted years associated with drought (Artaxo et al., 2013). Fires are widespread within the so-called 'arc of deforestation' which traverses the southern and eastern edges of the Amazon basin. This arc comprises both tropical forest rainforest and Cerrado (savannah-like) biomes which feature distinct fire regimes.

Previous studies have discussed the vertical distribution of biomass burning pollutants in TSA based on in situ aircraft measurements. The 1985 ABLE-2B study across northern Amazonia from the coast to Manaus provided profiles and transects of aerosol extinction (Andreae et al., 1988) , CO (Sachse et al., 1988), aerosol number and ozone (Gregory et al., 1988). The 1992 TRACE-A campaign in central and eastern TSA yielded vertical profiles of total aerosol and black carbon mass (Pereira et al., 1996). The 1995 SCAR-B and 2002 LBA-SMOCC studies focused on western TSA, providing vertical information on aerosol number (Andreae et al., 2004), aerosol scattering (Chand et al., 2006; Ross et al., 1998) and CO concentrations (Freitas et al., 2009). The 2004 TROFFEE campaign in central TSA provided vertical profiles of CO and $CO_2$ (Yokelson et al., 2007). Finally, the 2009 BARCA-A study provided aerosol number and CO concentrations in central and northern TSA (Andreae et al., 2012). Common among these studies is the limited regional spatial extent and the coverage above ~4 km, whilst many only show singular profiles and/or do not consider the eastern Cerrado fire regime. Furthermore, profiles are typically not integrated with multiple pollutants nor simultaneous thermodynamic measurements. This is important for understanding how mixing, advection and removal affect pollutants differently. Likewise, recent lidar remote sensing from satellite (Bourgeois et al., 2015; Huang et al., 2015) and surface (Baars et al., 2012) platforms do not provide the comprehensive dataset best suited to examining the drivers of multi-pollutant vertical distribution.

When simulating the pollutant vertical distribution over TSA there is a large diversity between AeroCom global models (Koffi et al., 2012) and between a subset of regional models (Andreae et al., 2012). In particular, open questions remain regarding treatment of convective mixing (Archer-Nicholls et al., 2015) and the injection of buoyant plumes into the vertical column (Paugam et al., 2016). Furthermore, to ensure consistency with remotely sensed AOD measurements, aerosol emissions

are typically up-scaled by a factor of two to five (Johnson et al., 2016; Reddington et al., 2016a). This represents a key uncertainty in model treatment of biomass burning aerosol, with deficiencies in both emissions inventories and model process posited.

In this study we examine the key controls on the vertical distribution of biomass burning pollution across TSA during the 2012 dry season using data collected on research flights during the SAMBBA (South AMerican Biomass Burning Analysis) field experiment. This study includes the first incandescence based measurements of black carbon mass and mixing state in a tropical biomass burning region, and also the most comprehensive integrated multi-parameter library of profiles for tropical South America. It also represents the first comprehensive airborne sampling in eastern Cerrado regions of intensive fire activity. To determine the key drivers of pollutant distribution we examine: features of the atmospheric structure (Sect. 3.1); their interaction with pollutants based on a novel and detailed analysis of individual profiles (Sect. 3.2); and, fire activity and horizontal aerosol distributions based on satellite observations (Sect. 3.3). Average pollutant profiles are presented across the different synoptic regimes over TSA (Sect. 3.4). Together, these approaches allow determination of the key drivers of the vertical distribution of biomass burning pollution, and an examination of how these manifest in different environments and relate to previous observations (Sect. 4). A discussion of implications and recommendations for future measurement and model approaches to reduce the uncertainty surrounding the impacts of biomass burning pollution both in TSA and other tropical biomass burning regions is presented (Sect. 5).

## 2 Experimental details

Twenty science flights were conducted during the SAMBBA experiment between 14[th] September and 3[rd] October 2012, primarily based out of Porto Velho in northern Rondônia state, Brazil (Fig. 1). Extensive geographical sampling was conducted via three basic flight operations: (i) straight and level runs within the boundary layer, (ii) high altitude surveys and (iii) profile ascents/descents. Flights were conducted by the UK BAe-146 research aircraft operated by FAAM (Facility for Airborne Atmospheric Measurements). The BAe-146 flew with a comprehensive instrumentation suite, capable of measuring aerosols, dynamics, cloud physics, chemical tracers and meteorological fields (McBeath, 2014). Simultaneously, a fully instrumented ground site was operational in Porto Velho, the results from which are described in Brito et al. (2014).

Aerosol total scattering coefficients were measured by a 3-wavelength integrating nephelometer (TSI Inc., USA; Anderson et al., 1996). Standard corrections were applied for angular truncation and non-Lambertian light source errors (Anderson and Ogren, 1998; Müller et al., 2011). The total scattering coefficient at 550 nm is reported at either dry RH (<30%), $\sigma_{sp\_dry}$, or ambient RH, $\sigma_{sp\_amb}$. These values were derived from the measured data based on the internal RH of the nephelometer and applying humidification scaling factors from Fig. 4c of Kotchenruther and Hobbs (1998). Note, however, there remains significant uncertainty on the role of water uptake on aerosol in TSA (Reddington et al., 2018, Darbyshire et al., in preparation).

Mass concentrations of refractory Black Carbon (rBC) were obtained using the Single Particle Soot Photometer (SP2, Droplet Measurement Technologies, USA; Baumgardner et al., (2004), Stephens et al., (2003)). The operation on board the

BAe-146 is outlined by McMeeking et al. (2010) and the calibrations and post processing undertaken for this work are described by Allan et al. (2014). Reported mass loadings have a measurement uncertainty of approximately 30% (Schwarz et al., 2008; Shiraiwa et al., 2008). The coating thickness of scattering material on core rBC particles was calculated following the method presented by Liu et al. (2014) and Taylor et al. (2015). By assuming a full concentric encapsulation of the spherical core, the coating thickness of single rBC particles are estimated using a core refractive index of 2.26-1.26i and coating refractive index of 1.50+0i (Taylor et al., 2015).

The SP2 and nephelometer sampled through a Rosemount inlet which has a high transmission for the submicron aerosols of interest here (Trembath, 2013). rBC and $\sigma_{sp}$ are reported at standard temperature (273.15 K) and pressure (1013.2 hPa) to allow direct comparison of particle composition at different altitudes.

Carbon dioxide mixing ratios were measured by a fast greenhouse gas analyser (FGGA; Los Gatos Research Inc.) adapted for aircraft use (O'Shea et al., 2013). The total accuracy of this measurement is estimated at ±0.17 ppm. Carbon monoxide mixing ratios were measured by an Aero-Laser AL5002 VUV resonance fluorescence gas analyser (Gerbig et al., 1999). Total uncertainty of the FAAM instrument is estimated to be ±2% (O'Shea et al., 2013). In flight calibrations were performed for CO and $CO_2$ using World Meteorological Organization traceable gas standards.

A General Eastern 1011B (GE Measurement & Control) chilled mirror hygrometer provided measurements of ambient dew point temperature, accurate to ±0.2 ◦C. A Rosemount/Goodrich type-102 True Air Temperature sensor was mounted outside the aircraft, providing ambient temperature measurements using a Rosemount 102AL platinum resistance immersion thermometer. A five hole radome-mounted turbulence probe at the aircraft nose provides measurement of airflow relative to the aircraft, thus allowing calculation of wind vectors when combined with a GPS inertial navigation unit (Petersen and Renfrew, 2009). A total of 56 Vaisala RD94 dropsondes were released from the aircraft at altitudes up to 8 km, providing geolocated profiles of pressure, temperature, relative humidity and wind vectors to ground level.

Individual profiles from the aircraft and dropsondes were averaged (median) onto a 50 m vertical grid for automated identification of thermodynamic features and their relation to haze distribution and pollutant plumes. Plumes identified on take-off/approach to an airport were removed from further analyses as they were likely contaminated from urban emissions.

Satellite remote sensing products were obtained for the SAMBBA period and September 2008-2017. The MODIS (Moderate Resolution Imaging Spectroradiometer) instrument provided retrievals of aerosol optical depth and active fire data. Precipitation rates were acquired from instruments on-board TRMM (Tropical Rainfall Measuring Missions). Land cover data was acquired from the ESA CCI Land Cover project and used the United Nations Land Cover Classification System. Specific details on the acquisition and processing of satellite data are provided in the supplement.

# 3 Results

## 3.1 Atmospheric structure

The evolution of the convective boundary layer is important for pollutant mixing, advection and plume injection and is idealised in Fig. 2. Surface heating initiates buoyant turbulent motions which deepen throughout the day and comprise the convective mixing layer. In the morning these thermals slowly destabilise the nocturnal stable layer before rapidly penetrating into the residual layer, i.e. the previous days' boundary layers. Growth of the mixed layer is inhibited by the statically stable entrainment zone, a region of overshooting thermals and downward entrainment from above. The lifting condensation level is typically within the entrainment zone but above the mixed layer top, leading to patchy cumulus fields. These clouds are either forced, i.e. fair weather cumulus which do not vent mixing layer air into the free troposphere, or active, where the level of free convection is reached and mixing layer air is vented through the tops of towering cumulus that may grow up to the limit of convection and detrain in the upper troposphere (Stull, 1985). Typically, a maximum in horizontal wind speed of variable magnitude and extent is present above the entrainment zone and referred to as the trade wind inversion.

The layer altitudes in Fig. 2 are based on a semi-automated analysis of individual thermodynamic profiles, the methodology for which is detailed in Sect. S2. This analysis is summarised for an example profile in Fig. 3A-C. From the wind speed and direction profiles (panel A), regions of local positive wind shear (vertical brown line) and locations of wind speed maxima (open purple circles) are identified. Profiles of relative humidity, specific humidity, potential temperature and equivalent potential temperature (panel B) act as the basis for identification of a mixed layer and where so, its depth. This involved a manual approach based on the spread of a number of automated methods. Profiles of temperature and dew point temperature (panel C) are presented in a skew-T log-P format, allowing calculation of the lifting condensation level (LCL), level of free convection (LFC) and limit of convection (LOC). The entrainment zone was not derived due to difficulties in determining its base with the information available. The format of Fig. 3 is replicated for each profile and displayed in Sect. S5.

In general, observed profiles of horizontal wind speed reflect those expected based on our understanding of synoptic flows over TSA (Campetella and Vera, 2002). In the lower troposphere (850 hPa) the mean synoptic circulation is an anticyclonic flow centred around south eastern TSA: easterly trade winds turn south-eastward toward the extra tropics, parallel to the Andes (Fig. 4C). Horizontal wind speed is greatest in eastern coastal regions, northern Amazonia and Bolivia. Slacker wind speeds are prevalent in south western TSA. At higher levels the influence of the Andes is reduced, resulting in a more zonal flow.

There is considerable structure in the observed profiles which is not captured by large scale wind fields, such as those in the ECMWF ERA-Interim reanalysis. This includes multiple filament like maxima in wind speed, often without accompanying changes in wind direction. In western TSA the expected sharp change in wind direction and increase in horizontal wind shear associated with a trade wind inversion is often not present and instead a more gradual change in speed and direction was observed. In the mixed layer, horizontal wind speed is typically lower and of a more uniform direction and magnitude. There are a number of exceptions to this generalisation, where positive wind shear regions or wind speed maxima are present through the mixed layer. A wind speed maximum is generally observed above or collocated with the mixed layer top (e.g. Fig. 3). At

times this was the trade wind inversion but typically it was not. If this feature is associated with a region of positive wind shear below, mixed layer growth is likely inhibited and in turn the magnitude of the jet and gradient of wind shear reduces.

Variations of the Fig. 2 idealised structure were also found to depend on the sample region, owing to the west-east asymmetries in land use. Compared to the Cerrado regions, the northern and western tropical forest regions experience higher

levels of precipitation, soil moisture and cloud fraction resulting in lower solar insolation and surface temperatures (Fig. S2). On average these resulted in a deeper, faster developing, mixing layer in the east than west and north (average of 1.8 km vs. 1.1 km). As such the lifting condensation level (2.6 km vs. 1.4 km) and level of free convection (3.1 km vs 2.4 km) was also greater. Only in 25% of profiles was the limit of convection identified, where it was higher in the east than west and north (4.4 km vs. 3.5 km).

Superimposed on the regional variation was the impact of two distinct meteorological phases in north and west TSA. Between 14-Sept and 22-Sept conditions were characteristic of the dry season, whilst between 23-Sept and 05-Oct conditions more closely resembled those of monsoonal transition (Brito et al., 2014). This second phase was characterised by an increased rate and area of rainfall, accompanied by reductions in solar insolation and increases in cloud fraction and soil moisture (Fig. 4). In ~30% of profiles no mixing layer could be identified and instead the lower atmosphere was conditionally unstable

throughout (and hence Fig. 2 is unrepresentative). This typically occurred in the west of TSA and toward the end of the campaign as wet season type conditions became more prevalent. Meteorological conditions during the 2012 SAMBBA period were typical of the climatology – there were no climatic extremes (Fig. S2).

## 3.2 Determining pollutant vertical distribution via feature analysis of individual vertical profiles

An automated analysis of interactions between atmospheric structure and pollutant abundance was undertaken for each

20 individual profile. Results from this analysis are summarised in Table S1 and illustrated for the example profile in Fig. 3D-G. The full analysis methodology is provided in Sect. S2.

When present, the convective mixing layer was investigated to determine if each pollutant was correspondingly well mixed throughout the layer. This is the case in approximately half of all profiles featuring a mixed layer. The profiles where this was not the case are typically perturbed by fresh plumes and/or wind shear and jet interactions. The number of CO and $CO_2$ profiles

taken during the morning that were classified as well mixed was approximately half that of the afternoon. For example in Fig. 3 - as indicated by the vertical green line and accompanying yes/no indicator - rBC and $\sigma_{sp\_dry}$ are considered well mixed but CO and $CO_2$ are not, due to concentrations increasing near the surface.

If pollutant abundance in the region above the mixing layer (or if absent, the near surface) and 4 km exceeded those of unpolluted background conditions, a pollutant residual layer was identified. Dry season background conditions were defined

as 0.1 µg m$^{-3}$ for rBC (Artaxo et al., 2013), 15 Mm$^{-1}$ for $\sigma_{sp}$ (Rizzo et al., 2013) and 140 ppb for CO (Andreae et al., 2012). Since $CO_2$ has significant global trends and the large biogenic sources/sinks its background is harder to quantify and is not considered here. Over 70% of profiles included a pollutant residual layer of rBC, $\sigma_{sp}$ and CO, even those in remote regions away from fresh emissions (Table S1). Typically, the shape of these residual layers is similar to that from the previous day(s)

boundary layer. However there is substantial diversity in shape and magnitude, especially between 2 - 4 km that result from variations in plume injection, long range transport, cloud detrainment or entrainment of clean air from aloft. This is seen in the profiles in Fig. 3 where the residual layer is indicated by the vertical brown line and accompanying yes/no indicator.

Two impacts of horizontal wind shear on pollutant vertical structure were explored. Firstly capping effects, indicated by a decline in pollutant abundance over regions of positive horizontal wind shear. This was present in 40-60% of instances. Secondly the curtailment by wind shear on plume rise via increased drag and lateral entrainment, as previously explored by Freitas et al. (2010). This was indicated by a co-location of pollutant abundance maxima with wind shear maxima. This was present in 20-30% of instances. Fig. 3 contains such an example at ~650 hPa, as illustrated by the purple circle in panel F.

Plumes were identified by a significant enhancement in pollutant abundance (dot-dash grey lines, Fig. 3.D-G) over a local background (dashed grey lines, Fig. 3.D-G). The majority of profiles contain a plume of one or more pollutant. Only in 10% of these plumes did all pollutant abundances exceed the identification threshold, despite often being well correlated, indicating variability in plume composition. The plumes observed in the mixing layer are likely more fresh and locally injected. Their frequency increases in the afternoon (Table S2) which may be associated with increased fire incidence (Giglio et al., 2006). Plumes were encountered between the mixing layer and lifting condensation level in 15-20% of profiles, indicating a significant number of fire plumes have sufficient energy to overcome the stable entrainment zone. If the pyro-convection is such that the plume reaches above the lifting condensation level, as in 10-25% of profiles, pyro-cumulus form and detrain pollutants. Plumes encountered at these levels likely include those from local sources and those advected from regional upwind sources. The frequency of these plumes generally declines in the afternoon indicating pollutants have been re-entrained into the mixing layer or, in western regions, transported following deep moist convection above our flight ceiling.

It is possible the layers of enhanced pollutants above the lifting condensation level are not direct plume injections but detrainment from active cumuli following shallow moist convection. However, mixing layer pollution transported in this manner will be of similar concentrations to that of the residual layer and as such may not be significant enough to trigger the plume thresholds defined here. Furthermore it would be expected that relative concentrations of CO would be conserved but those of aerosol would be reduced following activation and wet scavenging if precipitation occurred. However, analysis of the plume composition (Sect. 3.4) indicates the opposite to be true. Together this indicates the pollutant enhancements observed via this analysis are likely plumes rather than detrained layers. There is evidence of moist convection delivering CO to altitudes above ~4 km but with significant wet scavenging of aerosol. In 81% of profiles with sufficient vertical coverage CO loadings increased by 40 ppb from a minimum at ~4km to the top of the profile. Unlike the discrete signal from plumes the enhancement was often 1-2 km deep. Of the rBC and $\sigma_{sp}$ profiles only 8% and 3% had a similar increase in signal co-incident with CO enhancements. This indicates significant removal of aerosol from deeper convection to altitudes above ~4km.

### 3.3 Fire activity

During the dry season fires are prevalent across TSA but especially between 0 and 25°S (Fig. 5A). Pollutants emitted from these fires accumulate in south-western TSA as the Andes act as a barrier to smoke advected westward via the trade winds

(Fig. 5C). This regional smoke plume is optically thick and its shape and magnitude are dependent on fire activity in a given year. Whilst the September 2012 dry season is generally representative of years since 2008 (Fig. 5A-C ii.), fire count and radiative power were greater in the eastern states. This likely explains  an enhancement in AOD of ~0.1 in these regions and southern Mato Grosso state. The shift to wet season conditions corresponded to a reduction in fire count (Fig. 6A) and

emissions (Pereira et al., 2016). We speculate conditions were not optimal for human ignited fires, as a rise in relative humidity and increase in wind speed makes ignition and control more difficult, whilst decreasing the fuel consumption. The increase in cloud cover may have also reduced fire detection efficiency. Together with increased wet removal, these effects acted to decrease the regional AOD by ~0.2 and shift the peak AOD from central to north eastern regions (Fig. 6C).

## 3.4 Vertical distribution of pollution over synoptic scales

Based on the variability between the individual profiles and the regional differences in fire activity, land use and meteorology, we define four regimes which characterise the key synoptic differences in pollutant abundance and vertical distribution across TSA during SAMBBA:

−    Deforestation impacted western Amazonia around Rondônia and Mato Grosso states (68.5°W - 54°W, 12°S - 5.5°S) during dry season conditions (14-Sept to 22-Sept). This regime is henceforth referred to as W1.

−    The same area during dry-wet transition season conditions (23-Sept to 05-Oct). Henceforth W2.

−    'Pristine' rainforest area north of Manaus (65°W - 59°W, 5.5°S - 1°S) during the dry season on 19[th] Sept. Henceforth N1.

−    The dry Cerrado environment in eastern regions around Tocantins state (53°W - 46°W, 12°S - 9°S) between 26[th] Sept. and 2[nd] Oct. Henceforth E0.

The geographic extent of these regimes is demonstrated in Fig. 1 and the median thermodynamic variables in Fig. 7.

The average aerosol vertical profile shapes of W1, W2 and N1 are similar (Fig. 8.A,B) - similar loadings occur from the near surface to 3 km with a small maximum between 1 and 2 km. Above 4 km, loadings decline to near background values. Comparatively, E0 is characterised by a near surface maximum, which may be because multiple fresh fires were sampled, followed by similar loadings up to 4 km, above which concentrations declined sharply to near zero. The shape of the aerosol profiles is not significantly affected if plumes are removed from the analysis.

There are also contrasts in the aerosol abundance between the regimes. There is a twofold reduction throughout the column between W1 and W2, consistent with the satellite AOD fields (Fig. 6). Within the boundary layer this represents a reduction from 0.85 to 0.4 µg m$^{-3}$ for rBC and 115 to 60 Mm$^{-1}$ for $\sigma_{sp\_dry}$. rBC loadings of 0.85 µg m$^{-3}$ in N1 are similar to those in W1, yet $\sigma_{sp\_dry}$ of 70 Mm$^{-1}$ are more akin to W2. E0 features a similar ratio between rBC and $\sigma_{sp\_dry}$ as N1, although the abundance is much larger at 1.7 µg m$^{-3}$ and 125 Mm$^{-1}$.

For all regimes CO mixing ratios are greatest at the near surface: 340 ppb in W1, 310 ppb in E0, 220 ppb in W2 and 150 ppb in N1. CO mixing ratios reduce with altitude to a minimum in the free troposphere: 125 ppb at an unknown altitude in W1, 95 ppb at 4-5 km in E0, 150 ppb at 4-5 km in W2, 100 ppb at 3-4 km in N1. The reduction of CO mixing ratios in the boundary layer from W1 to W2 corresponds to that of aerosol abundance. $CO_2$ mixing ratios are also greatest at the near

surface: 402 ppm in E0, 393 ppm in N1, 394 ppm in W1 and 397 ppm in W2 representing a particularly prominent enhancement. $CO_2$ mixing ratios also reduce with altitude with minima of 391 ppm in E0 and N1 or 392 ppm in W1 and W2 at the same altitudes as those for CO.

Profiles of pollutant ratios further illustrate the contrasting aerosol and gas phase vertical distributions (Fig. 9.A-B). Compared with CO and $CO_2$ aerosol abundance is enhanced at higher altitudes in the boundary layer, even accounting for the removal of plumes (Fig. S3). The peak enhancement is at ~2 km in W2, ~1.5 km in N1, ~4 km in E0 and, in W1, ~2.5 km for $\sigma_{sp\_dry}$ and ~1.5 km for rBC. These distributions are also seen when examining the pollutant ratios within plumes - as altitude increases so does the plume Δaerosol:ΔCO ratio (Fig. 9.C-D). We use the term Δaerosol:ΔCO to describe both ΔrBC:ΔCO and Δ $\sigma_{sp\_dry}$:ΔCO since these aerosol properties co-vary. As these enhancement ratios are typically within the boundary layer, share a common source and in most cases are likely relatively fresh, it is unlikely they are bias by sudden changes in the composition of background air driving the observed gradient, as warned against in Yokelson et al. (2013).

There is limited evidence for a diurnal variation in aerosol abundance and vertical profile shape. CO, however, is enhanced near the surface in the morning (Fig. 8). In W2 and N1, $CO_2$ is also enhanced at the near surface in the morning but, given the poor correlation with other pollutants (Sect S5), likely arises via biogenic activity. This may also help explain the greater $CO_2$ mixing ratios in W2 vs. W1, as the increased cloud cover reduces photosynthesis and hence $CO_2$ uptake (Graham et al., 2003).

Within the boundary layer there is a significant contrast in the coating thickness of rBC between E0 (55 nm) and W1, W2 and N1 (80-90 nm; Fig. 10, Fig. S4). Coating thickness increases with altitude in the boundary layers of W2, N1 and E0. This may be associated with partitioning of organic material into the particle phase as temperatures decrease, and/or the greater prevalence of fresh emissions near surface. In W1, W2 and most markedly N1, the rBC coating thickness declines above the boundary layer. Comparatively, rBC coating thickness increases above the boundary layer in E0.

## 4 Discussion

### 4.1 Drivers of the pollutant vertical distribution

Fire class determines the relative mix of pollutants emitted. In E0 there are relatively greater loadings of rBC owing to emissions from more efficient flaming combustion in Cerrado fires as observed by Hodgson et al. (2018). In W1 and W2 there are relatively greater magnitudes of $\sigma_{sp\_dry}$ and CO as the higher biomass density and moisture content results in more inefficient smouldering combustion. Significantly, the shift in meteorology between these two phases does not substantially impact the relative abundances of rBC, $\sigma_{sp\_dry}$ and CO to each other.

Meteorological conditions influence the absolute pollutant concentrations via their impact on removal rates/processes, advection and fire incidence. The driest conditions and greatest fire incidence meant the greatest pollutant concentrations were observed in E0. The reduction in aerosol loadings from W1 to W2 is attributable to a corresponding decline in fire count and increase in wet removal associated with the more widespread and heavy precipitation. Aside from wet removal, the ubiquity of the pollutant residual layer indicates pollutants are not rapidly removed. As such, advection becomes a key modulator of

pollutant concentrations local to, and distant from, the source region. The latter is the case in N1 - back trajectory analysis (Fig. S6) shows that pollutants are advected from the fire hotspot around Maranhão and northern Pará states (Fig. 5). The pollutant mix - relatively greater concentrations of rBC and – is consistent with the Cerrado burns at these sources. Whilst $CO_2$ mixing ratios are also greater in the east (relative to CO), as would be expected from more flaming combustion, the role of fire combustion processes cannot be evaluated given the confounding influence from biogenic sources/sinks in both regions.

The shapes of pollutant vertical distributions are primarily controlled by meteorological conditions, in particular vertical convective motions and horizontal wind shear (Fig. 7). The former acts to mix pollutants released near the surface toward the mixing layer top, the altitude of which can be modulated by the latter, soil moisture and solar insolation. The difference in profile shape from west to east to north is primarily driven by contrasting mixed layer depths. Pollutant loadings remained relatively high above the mixing layer in residual layers, indicating wet removal is not significant at these altitudes. Large unmixed plumes perturbed the mixed and residual layers , although they contribute only 15% (E0), 11% (W1), 8% (W2), and 1% (N1) to the scattering only column AOD (calculated from the nephelometer, Sect. S2). Such plumes were seldom seen above 4 km, in contrast to previous observations in approximately the same sample region (Supplementary material in Andreae et al., 2004), indicating the mass flux from large pyrocumulus detrainment into the upper troposphere (within the aircraft range) was not significant. The observed increases in CO concentrations above ~4km indicate vertical transport of mixing layer pollution into the free troposphere. The presence of co-incident increases in rBC or $\sigma_{sp\_dry}$ in less than 10% of these plumes indicates moist deep convection transports CO and presumably other gaseous and non-soluble components to altitudes greater than 4 km but efficiently removes aerosol from the air by wet scavenging. Such efficient aerosol removal and transport of CO to the upper troposphere was also observed in TSA during the 2014 dry season by Andreae et al. (2018). This is consistent with the decrease in rBC coating thickness at these altitudes in W1, W2 and N1 and also is similar to previous observations in Boreal Canada that showed preferential wet deposition of the largest and most coated particles (Taylor et al., 2014). As deep moist convection is not common in eastern regions (e.g. using TRMM rainfall as a proxy, Fig. 4a) the source of elevated and enhanced CO is unlikely to arise from the mixing layer in the east. It is possible the source is from CO aloft in the west which is recirculated in the persistent anti-cyclonic flow at 500 hPa (Fig. S2.f) as has previously been observed from satellite (MOPPIT) CO observations by Deeter et al. (2018) and characterised by trajectory analysis in Andreae et al. (2018). A long aging time is consistent with the larger rBC coatings observed aloft in E0.

A transect flight from east to west (Fig. 11) captures and summarises the meteorological drivers of the regional contrast in pollutant vertical distribution. A declining mixing layer depth from east to west is evident from the reduction in altitude of the sharp gradient (i.e. the entrainment zone) of $\theta_e$ from ~3 km in the east to ~1.5 km in the west. Above the mixed layer top relative humidity increases, especially so above the lifting condensation level where the high humidity distinguishes the cloud convective layer. This layer is deeper in the west, associated with deeper moist convection. A wind speed maximum is present at 5-6 km, coincident with the entrainment zone. Together, this structure can explain the lidar derived extinction coefficient distribution (first published by Marenco et al., 2016). Aerosol is capped below the first wind speed jet, well mixed within the mixed layer and features a maximum at ~1.5-2 km. Visible plumes at ~59 °W and ~52.5 °W lie at injection heights typical of

those observed in the in situ profile data. The similarity of the in situ $\sigma_{sp\_amb}$ and lidar extinction coefficient profile shapes at the regional (Fig. 11) and local (Fig. S7) scales engenders confidence in the representativity of both datasets. Disparity in the absolute magnitudes is primarily explainable by differences in the sampling coverage (Fig. S8).

Consistent values of the scattering angstrom exponent at 700/550 nm within the boundary layer (Fig. S5) indicate a similar aerosol type throughout. These values were all above 1.5, typical of submicron biomass burning aerosol and indicating no significant regional role for super-micron dust or primary biological aerosols which have values closer to zero (Clarke et al., 2007; Russell et al., 2010). This is consistent with size distributions reported and from the aircraft (Darbyshire et al., in preparation) and literature values of the angstrom exponent during the dry season (Rizzo et al., 2013; Saturno et al., 2018). This also indicates that model scaling factors to match remotely sensed AOD are not significantly biased by non-biomass burning aerosol. The observed increase in humidity with altitude in the boundary layer (Fig. 7C) will likely have a significant impact on AOD and therefore the required scaling factor as it remains an uncertain model process (Johnson et al., 2016; Reddington et al., 2018). We note that findings from SAMBBA suggest that omission of burned area from small undetected fires is the most significant source of inventory under-representation of aerosol emissions (Hodgson et al., 2018; Reddington et al., 2016b), as in other burning regions (Nowell et al., 2018).

The average vertical distribution of pollutants in this study are similar to those from previous in situ and remote sensing observations in the region, despite temporal, geographic and instrumental sampling differences. This indicates driving processes are common throughout. The aerosol vertical profile is approximately vertical from the surface to a height of: 2.25 km (Huang et al., 2015; Table 2 therein), ~1.5 km (Bourgeois et al., 2015; Fig. 6 therein), ~2 km (Baars et al., 2012; Fig. 12 therein), ~1.5 km (Andreae et al., 2012; Fig. 6 therein), ~2 km (Marenco et al., 2016; Fig. 9 therein) or 1.5 - 3.5 km (this work, Fig. 8). Profiles of rBC in Andreae et al, (2018) were similar to those presented here – showing greater concentrations in Cerrado regions where lofted layers of greater concentrations were also observed between 2-4 km. Loadings reduce to near baseline values above ~4 km in all. Only in Andreae et al. (2012) have similar regionally averaged profiles of CO been presented. These exhibit a similar profile shape as here - greatest near the surface and reducing thereafter to a minima at ~4 km. Andreae et al. (2012)therefore also demonstrated the same relationship of Δaerosol:ΔCO with altitude as observed here. Whilst studies displaying only a few profiles (e.g. Andreae et al., 1988, 2004; Chand et al., 2006; Pereira et al., 1996; Yokelson et al., 2007) may not mirror these average distributions, comparable profiles are present in our library of individual profiles (Sect S5) indicating that our more extensive dataset is consistent with previous reports and captures variability between different studies.

To try understand the relationship between Δaerosol:ΔCO and altitude it is necessary to examine multiple potential drivers. Substantial dry deposition could remove near surface aerosol and reduce the observed Δaerosol:ΔCO. However, dry deposition fluxes reported in the literature (Ahlm et al., 2010) are small compared to the number concentrations observed (Darbyshire et al., in prep) so this is an unlikely cause. Likewise we discount a significant anthropogenic non-biomass burning source of CO near the surface, as emissions hotspots are over 1000 km from the flight region and are approximately an order of magnitude lower than from fires (Fig. S9). Increasing aerosol concentrations with altitude are typically attributed to partitioning of organic

and inorganic species from lower volatility gas phase species into the particulate phase as temperatures decline and relative humidity increases (Heald et al., 2011; Morgan et al., 2010) . A significant fraction of boundary layer organic aerosol in the regional haze will be secondary in nature following oxygenation and condensation of semi-volatile vapours during smoke plume evolution. During SAMBBA no net addition of organic mass was observed in the near or far field (Morgan et al., 2019), unlike previous studies (e.g. Akagi et al. 2012). However, the contribution of further biomass burning secondary organic aerosol owing to increasing altitude is difficult to determine with the data available. There is an increase with altitude of the contribution from secondary inorganic aerosol with altitude in E0, but not W1,W2 or N1 (Darbyshire et al., in prep). This may account for the increase in rBC coating thickness with altitude (Fig. 10), but the decrease with altitude in W1, W2 and N1 indicates no significant addition of secondary organic or inorganic aerosol. Furthermore, as the approximate three-fold increase of involatile $\Delta rBC:\Delta CO$ with altitude is of the same magnitude to that of $\Delta\sigma sp:\Delta CO$, these secondary aerosol processes cannot explain our observations.

Plume $\Delta aerosol:\Delta CO$ enhancement ratios increase with altitude within the boundary layer, indicating that these plumes may be the source of the observed $\Delta aerosol:\Delta CO$ profiles in the regional haze. Past measurements (Fig. S10; Ferek et al. 1998; Yokelson et al. 2007) and those from SAMBBA (Fig. S11, Hodgson et al. 2018) indicate $\Delta aerosol:\Delta CO$ increases with MCE. We speculate the distribution may be driven by the relationship between MCE and fire intensity – and thus by extension, when normalising for meteorology, plume height (Lavoué et al., 2000). The peak heat flux (i.e. intensity) of an open landscape fire occurs when there is both a flame front, where flaming combustion occurs (high MCE), and zone of smouldering combustion (moderate MCE) in its wake. Once the flame front is extinct the fuel bed remains smouldering (moderate MCE) and although reduced, there may still be sufficient heat flux to generate a convective column to loft these emissions, albeit to intermediate altitudes. Once the heat flux is insufficient to generate any convective plumes and emissions from residual smouldering (low MCE) are released at the surface. Together one would thus expect a continuum from high altitude release of high+moderate MCE emissions to surface release of low MCE emissions, which could explain our $\Delta aerosol:\Delta CO$ observations. The evolution of plume height as a function of heat flux has previously been observed for savannah fires in South Africa (Stocks et al., 1996). Alternatively, there may be multiple convective cores for a given burn as seen elsewhere (Achtemeier et al., 2011). Where these are away from the primary flame front the MCE will be lower along with heat flux and therefore injection height. Another complicating factor may be from very intense fires which experience oxygen deficiency in the flaming zone and combustion efficiency drops (Ward and Hardy, 1991) – and therefore aerosol:CO reduces. We recommend further fire-scale plume dynamics observations in tropical regions to test these speculations and acknowledge that many case studies would be required to account for the other impacts on 'real world' fires plume rise (including stability, vertical profile of wind speed/direction, latent heat release, flame front annihilation, topography).    In addition to being related at the individual fire level, regionally there is a connection between MCE, average fire intensity and injection height. Fires in eastern Cerrado regions have a higher MCE (Hodgson et al., 2018), are more intense as illustrated by the FRP distributions in Fig. 5b and Gonzalez-Alonso et al., (2019), and have higher plume injection heights in-spite of more stable conditions (Fig. S12, Gonzalez-Alonso et al. 2019; Marenco et al. 2016).

Long range transport of emissions from biomass burning in tropical regions of West Africa may provide an alternate explanation for, or provide an additional contribution to, the increase in aerosol:CO with altitude. Such a transport pathway has long been observed (Andreae et al., 1994; Baars et al., 2011; Das et al., 2017) and typically peaks between August and October (Saturno et al., 2018). A basic investigation of CALIPSO Lidar retrievals in the air-mass history footprints of E0 (Fig. S13), shows frequent - but not persistent - offshore layers of enhanced backscattering between 2-5 km, identified as 'Elevated Smoke' by the aerosol subtype algorithm. These aerosol extinction coefficient of these layers would account for approximately half of the burden observed in E0. This is consistent with satellite AOD observations on and offshore from September 2012 (Saturno et al., 2018). These elevated smoke layers can be tracked back across the Atlantic to West Africa, where the elevated layers are typically deeper, more backscattering and geographically wider. If the aerosol:CO and BC:CO ratio in these layers is higher than that of emissions in TSA then introduction of this pollution could well generate the aerosol:CO gradient observed. As current literature estimates of the combustion efficiency and ratio between emission factors of rBC and CO are similar for both regions (Hodgson et al.,2018, and references therein) one may discount this theory. However, observations from recent (and as yet unpublished) field projects by the research group behind this study show greater rBC:CO enhancement ratios in lofted outflow from West Africa than over TSA. Furthermore, during the 2014 ACRIDICON-CHUVA aircraft campaign in TSA (Wendisch et al., 2016), offshore pollutant layers between 3-4 km with air mass histories from West Africa had rBC:CO enhancement ratios of approximately 20 and rBC loadings of up to 2 $\mu g \ m^{-3}$ (M. Andreae, personal communication). It remains unclear if these lofted layers with high rBC:CO enhancement ratios are driven by a difference in fire intensity or perhaps result from the MCE-plume dynamics mechanism speculated above, which would be consistent with past literature values.

## 4.2 Implications

We have observed a marked contrast in pollutant composition and vertical distribution from west to east across tropical South America which has not previously been identified. Given predicted 'savannisation' of the southern and eastern edges of the Amazon rainforest owing to climate change, deforestation and fire-climate feedbacks (Nobre et al., 2016) we recommend model analyses are carried out to assess the magnitude of these effects which may impact convective motions, cloud formation and regional dynamics.

The linkages between pollutants and atmospheric structure presented here reinforce the first order requirement for representation of realistic thermodynamics within models. This is currently a challenge for regional and global models, especially sub-grid convective mixing which drives the aerosol profile shape (Archer-Nicholls et al., 2016; Hong and Dudhia, 2012). For instance ECMWF ERA-Interim equivalent potential temperature profiles compare poorly to our measurements (Fig. 11) and those of Beck et al. (2013). These model fields are often used as boundary conditions for regional models and as such may introduce unrealistic thermodynamic structures and thus pollutant vertical distributions. Whilst the broad shape of horizontal wind shear from model output is captured on the individual profile basis, the fine structure present in observations is missing. Comparisons with ECMWF ERA-Interim and WRF output indicate this is the case for both global models and finer

scale regional models. This may be problematic in accurately representing sub-grid processes, such as plume rise and convective mixing.

We speculate the Δaerosol:ΔCO profile may be driven by a coupling between combustion efficiency and plume dynamics. If confirmed by further enquiry then future modelling studies will have to consider how best to represent the phenomenon. Replicating the gradient Δaerosol:ΔCO may be particularly important for model simulations which draw results based on a realistic vertical distribution. For example aerosol-cloud and aerosol-radiation interactions, or surface air quality simulations. Emissions factors for residual smouldering combustion in the Amazon have been collected (Bertschi et al., 2003; Christian et al., 2007), and could be used to test predictions from novel (and non-trivial) model set-ups against surface and satellite observations. Past satellite observations have suggested residual smouldering is a large source of CO and not fully captured by emissions inventories (Deeter et al., 2016; Pechony et al., 2013). If emissions from higher MCE combustion are released at altitude, further exploration of the processes and variability in TSA is required. Archer-Nicholls et al. (2015) demonstrated how the parameterised Freitas et al. (2007) plume rise model calculated injection heights which were too high in TSA. Climatological distributions of plume injection heights, which can prescribe model injection heights, show substantial diversity in TSA. The distributions derived by Marenco et al. (2016), consistent with plume heights here (Fig. S10), show injection heights at greater altitudes than those from CALIOP (Sofiev et al., 2013), and in turn MISR (Gonzalez-Alonso et al., 2019).

Although the magnitude and bearing may differ, the fundamental drivers of the pollutant vertical distribution identified here will remain so in drought years, which may be increasing in frequency (Jiménez-Muñoz et al., 2016). Dry convection may be more vigorous and the atmosphere more stable, deep convection less vigorous and aerosol scavenging reduced, fires more intense and fire hotspots located in different regions, but so long as model simulations well represent the fundamental drivers identified in this work then they ought to be able to replicate the resultant vertical distribution. This is a promising avenue for future research to predict the impacts in future years, following on from the study of Thornhill et al., (2018).

## 6 Conclusions

The vertical distribution of biomass burning pollutants regulates their subsequent horizontal distribution and thus the magnitude of associated impacts on weather, climate and air quality. In tropical biomass burning regions, pollutant vertical distributions and their driving processes are poorly characterised. Here, a novel integrated analysis of individual profiles from research flights in tropical South America will aid future experiments in reducing this uncertainty.

The thermodynamic structure of the lower troposphere was found to be critical in determining the pollutant vertical distribution on the local and regional scale. Pollutants were typically confined to the atmospheric boundary layer in either an active or residual convective mixed layer. This was deeper in the east (~3 km) than west (~1.5 km) reflecting a regional contrast in soil moisture and surface insolation and as such significant differences in sensible and latent heat. Above the boundary layer, enhanced concentrations of CO were likely transported via deep moist convection which removed aerosols via wet scavenging. Horizontal wind shear was important in advecting pollutants, inhibiting mixed layer growth and the vertical ascent of smoke

plumes from fires. In order to accurately simulate the vertical and regional distribution of biomass burning pollution our results highlight not only the importance of capturing boundary layer dynamics and convection adequately, but also the release of pollutant plumes at altitude.

During dry season conditions we observed a significant contrast in the pollutant haze composition between the western and eastern regions, which corresponded to different fire regimes. Whilst aerosol scattering coefficients of 120 $Mm^{-1}$ and CO mixing ratios of ~310 ppb were similar, black carbon mass loadings were much greater in the east than west - 1.5 vs. 0.85 µg $m^{-3}$. Cerrado burns in the east are more flaming, whilst those in the west, of primary/secondary forest and pasture land are more smouldering (Akagi et al., 2011; Andreae and Merlet, 2001; Hodgson et al., 2018). The regional contrast in biomass burning emissions resulting from these different fire types clearly merits future investigation in modelling studies to assess potential impacts on aerosol optical properties and their radiative effects.

Following the transition to wet season conditions in the west, fire activity declined and the observed concentration of black carbon mass, aerosol scattering coefficients and CO mixing ratios decreased. However, the ratios between these pollutants remained similar to ratios observed in the drier phase indicating little change to the composition of fire emissions despite the shift in meteorological conditions.

In the remote rainforest region of northern Amazonas state, significant concentrations of well mixed and thickly coated black carbon particles were observed in the lowermost 2 km and with comparatively low aerosol scattering coefficients. Trajectory analysis indicated origins from north-east Brazil where Cerrado fires were prevalent and, based on observations of Cerrado fires elsewhere, likely emit aerosol with a relatively high BC mass fractions. Given the high fire count and fire radiative power in this source fire region and the prevailing low level easterlies, the export of absorbing aerosol into the remote central regions of Amazonia may commonly occur during the dry season.

In all regimes we observed an increase of aerosol abundance relative to CO with altitude both in the background haze and plume enhancement ratios. This is unlikely associated with thermodynamic partitioning, as the phenomenon is observed for involatile rBC, nor surface deposition or non-biomass burning sources. Aerosol abundance relative to CO also varies with combustion efficiency, and we speculate that this may drive the observed gradient as combustion efficiency and plume injection height can be closely coupled. Alternatively, or additionally, the transportation of flaming rich emissions from biomass burning in West Africa at 3-4 km may drive the gradient. If the observed gradient is not fully captured by model simulations this may impede accurate air quality forecasts and predictions of aerosol-radiation or aerosol-cloud interactions. Further enquiry is recommended to fully explain these observations and explore the ramifications for regional climate and air quality.

The results here are likely applicable to other tropical biomass burning regions where observations of processes affecting pollutant vertical distributions are limited. Given the important impacts of biomass burning on meteorology, climate, air quality and ecosystem services, further vertically resolved observations of aerosol and trace gas pollutants are recommended.

**Data Availability**

All raw time series data used to derive the vertical profiles from the FAAM research aircraft are publicly available from the Centre for Environmental Data Analysis website (http://www.ceda.ac.uk/, last access: 31 August 2018). A direct links to the flight data records are given in the reference list (Facility for Airborne Atmospheric Measurements, Natural Environment Research Council, and Met Office, 2014). Raw active fire and Land use data used in the manuscript are available publicly from NASA and ESA respectively (see Acknowledgements). Processed individual and averaged vertical profiles, data masks, plume composition, model output and satellite fields and are currently available on request from E. Darbyshire. Lidar data is available on request from F. Marenco (franco.marenco@metoffice.gov.uk).

**Author Contributions**

ED analysed the data and wrote the manuscript with the aid of WTM, HC, BJ, JFB and FM. WTM, JDA, DLiu, DLowe, SO'S, FM and KS provided additional data analysis support, including data processing and quality assurance. MJF, JD, KS and JFB operated aerosol instruments during the field campaign. SB operated gas phase instruments during the field campaign. BJ, JMH, KML, PA and HC led the planning of the field campaign and were co-principal investigators on the SAMBBA project.

**Competing Interests**

The authors declare that they have no conflict of interest.

**Special Issue Statement**

This article is part of the special issue "South American Biomass Burning Analysis (SAMBBA)". It is not associated with a conference.

**Acknowledgements**

We would like to thank those involved in the SAMBBA project. This includes the Facility for Airborne Atmospheric Measurement (FAAM) and DirectFlight Ltd. who manage and operate the BAe-146-301 Atmospheric Research Aircraft (ARA), which is jointly funded by the Natural Environment Research Council (NERC) and the Met Office. A number of institutions were involved in logistics, planning and support for the SAMBBA campaign: The Met Office, INPE, University of Sao Paulo and the Brazilian Ministry of Science and Technology. Moderate Resolution Imaging Spectroradiometer (MODIS) AOD data (MOD08_D3.051) and Tropical Rainfall Measuring Mission (TRMM) precipitation data were retrieved

from the GIOVANNI online data system (http://disc.sci.gsfc.nasa.gov/giovanni), developed and maintained by the NASA GES DISC. Active fire data was produced by the University of Maryland and acquired from the online Fire Information for Resource Management System (FIRMS; https://earthdata.nasa.gov/data/near-real-time-data/firms/abouts; specific product: MCD14ML). Land cover data was provided to the United Nations (UN) Land Cover Classification System by the ESA CCI

Land Cover project (https://www.esa-landcover-cci.org/). ERA-Interim soil moisture, relative humidity, wind fields and temperature data was provided courtesy of ECMWF. We acknowledge those involved in the Multi-angle Imaging SpectroRadiometer (MISR) Plume Height Project for analysis and provision of data. The lead author was supported by a NERC studentship NE/J500057/1 and NE/K500859/1, and the SAMBBA project by NERC grant NE/J010073/1.

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

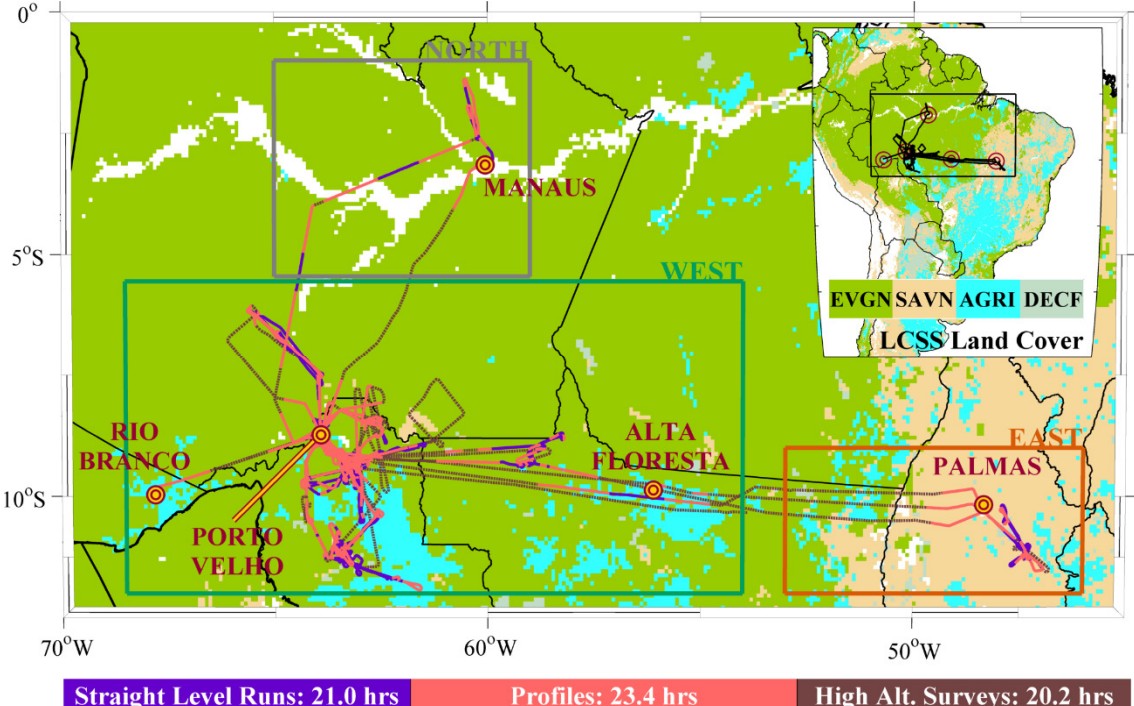

**Figure 1. SAMBBA operational domain with flight tracks coloured by basic aircraft operation. 2012 land use based on a simplification of the UN Land Cover Classification System (LCCS) classification as described in the supplementary information: evergreen forest (EVGN), savanna (SAVN), deciduous forest (DECF) and agricultural land (AGRI).**

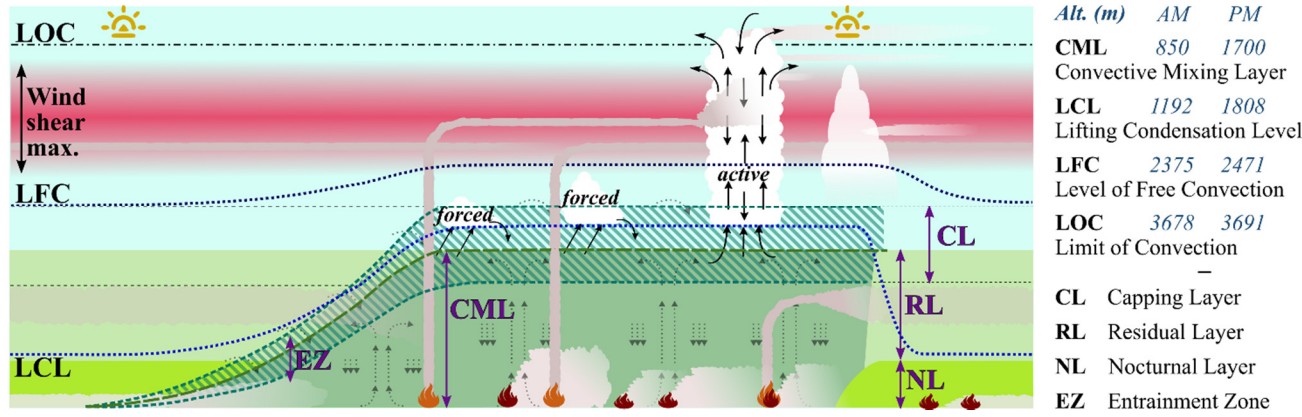

**Figure 2. Schematic of the typical diurnal development of the convective boundary layer. This is based on visual and automated analysis of individual profiles and, for the nocturnal period outside our sampling times, previous tropical boundary layer literature. Red flame symbols indicate more smouldering fires and orange symbols more flaming combustion.**

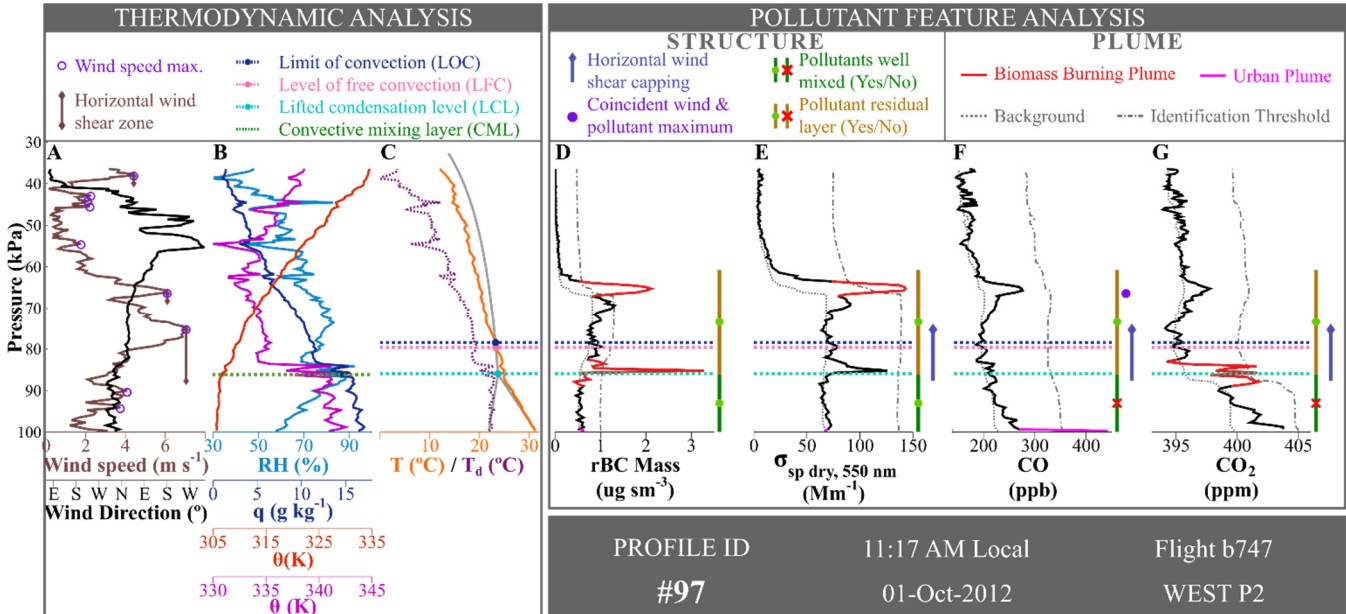

**Figure 3. Individual profile #97, from flight b747 (01/10/2012) near Porto Velho in Rondônia state. From left to right are the vertical profiles of (A) wind speed and direction, (B) relative humidity (RH), specific humidity (q), potential temperature (θ) and equivalent potential temperature (θe), (C) temperature (T) and dew point temperature (T_d) on a skew-T log-P scale, (D) refractory black carbon (rBC) mass, (E) aerosol scattering coefficient (550 nm), (F) CO and (G) CO₂ mixing ratios. Thermodynamic features are illustrated in panels A-C (see Sect. 3.1). Pollutant feature analysis is illustrated in panels D-G (see Sect. 3.2). Note, the red outline of a biomass burning plume is only present if it was identified for that specific species. For example at 65 kPa rBC and σ_sp_dry pass the identification threshold (grey dot-dash line) but CO and CO₂ do not.**

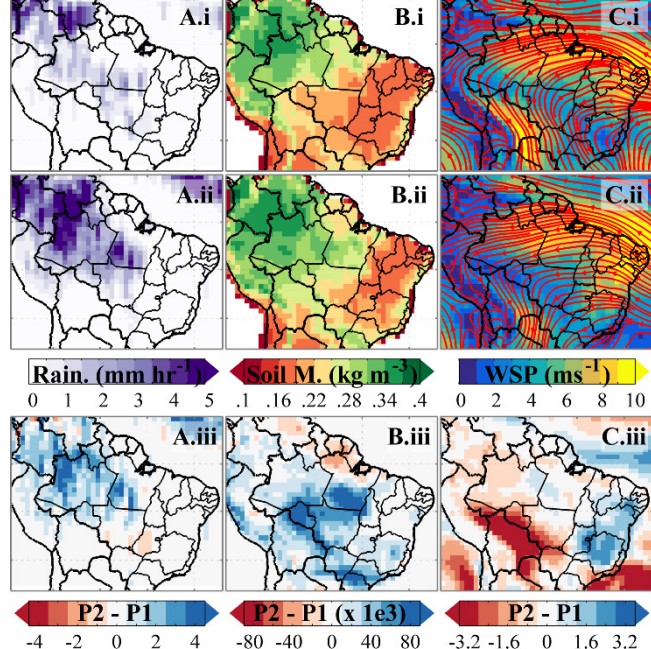

**Figure 4. Precipitation (A), soil moisture (B) and 850 hPa wind speed and flowlines (C) during meteorological phases 1 (i, dry season) and 2 (ii, dry-wet transition season) and the difference between the two periods (iii). Precipitation is derived from TRMM measurements. The soil moisture and wind speed products are derived from ECMWF Era-Interim product.**

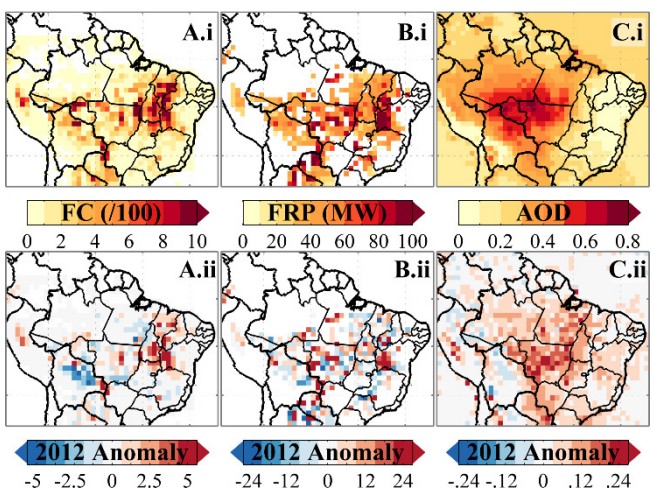

**Figure 5. September 2008-2017 median total fire count (FC; A) and fire radiative power (FRP; B) from MODIS active fire detections, calculated over a 1° grid. The FC scale is divided by 100 for figure clarity. September 2008-2017 median aerosol optical depth (AOD; C) from MODIS 1° product. The 2012 anomaly for each of these fields is illustrated in panels (ii).**

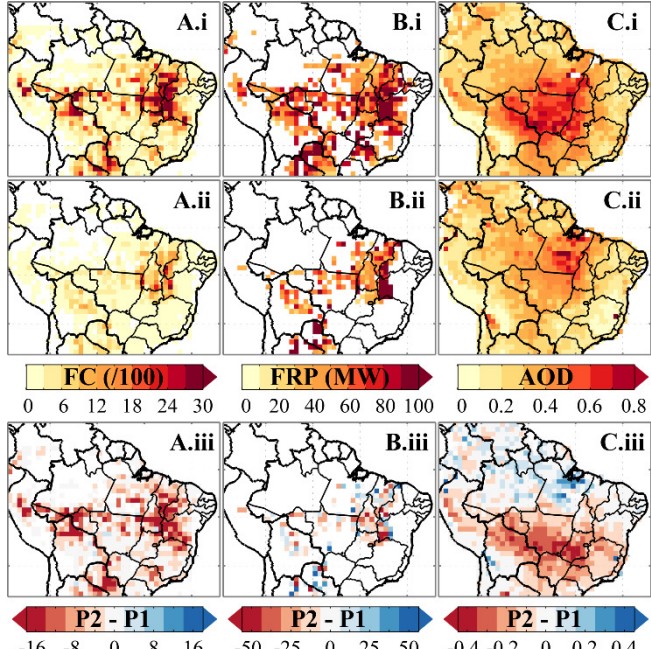

10    **Figure 6. Meteorological phases 1 (i; dry season 14th-22nd Sept 2012) and 2 (ii; dry-wet transition 23rd-5th Oct 2012) median total fire count (FC; A) and fire radiative power (FRP; B) from MODIS active fire detections, calculated over a 1° grid. Meteorological phases 1 (i; dry season) and 2 (ii; dry-wet transition) median aerosol optical depth (AOD; C) from MODIS 1° product. The difference between these periods is illustrated in panels (iii).**

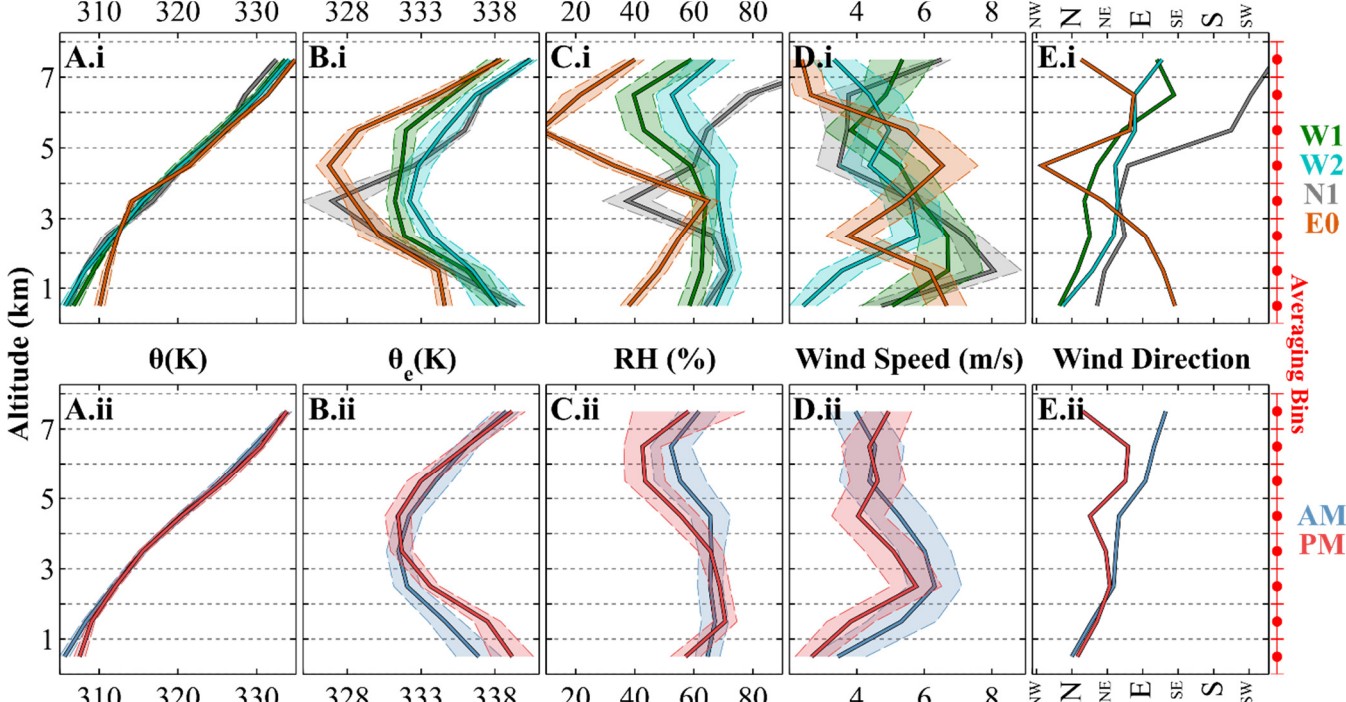

**Figure 7.** Median pollutant profiles averaged over the different regimes (top panels) and time of day (bottom panels, includes data from all regimes). Lighter shading represents the median absolute deviation.

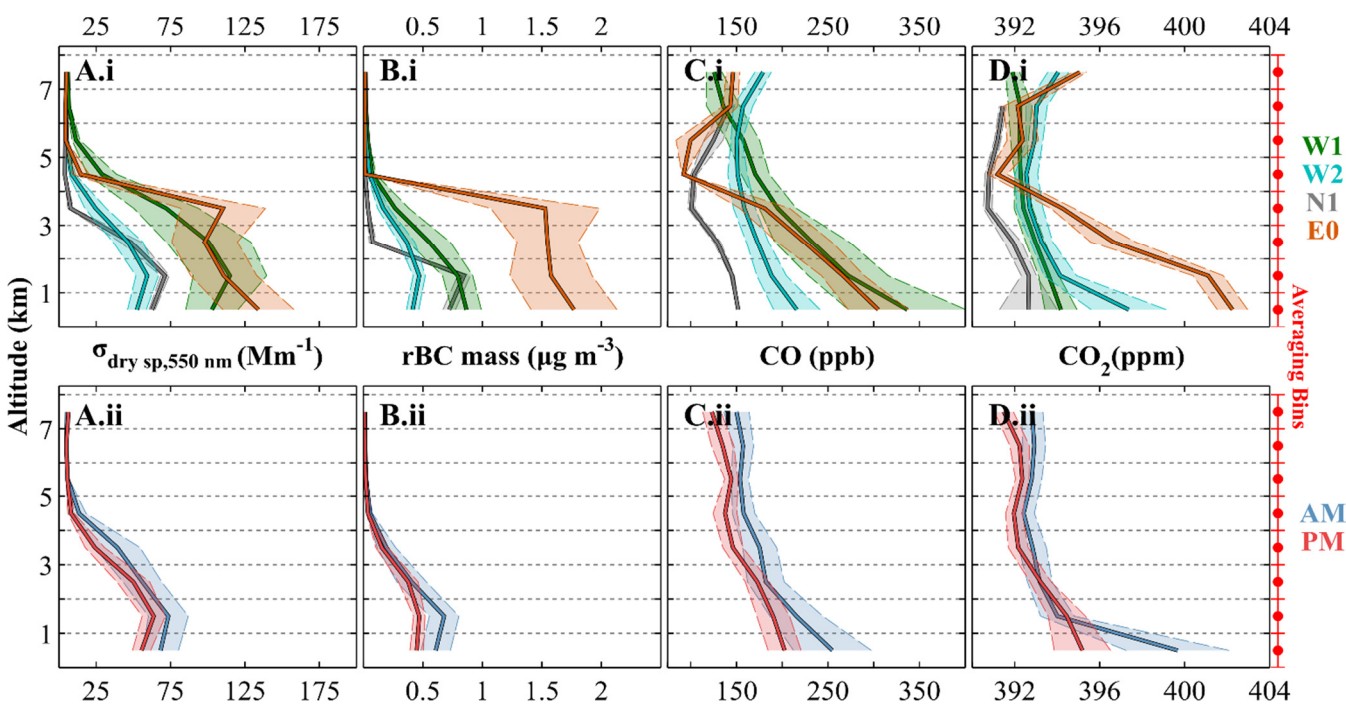

**Figure 8.** Median pollutant profiles averaged over the different regimes (top panels) and time of day (bottom panels, includes data from all regimes). Lighter shading represents the median absolute deviation. Note, $\sigma_{sp\_dry}$ is reported at standard temperature, standard pressure and 30% RH and hence the column AOD cannot be derived from these profiles.

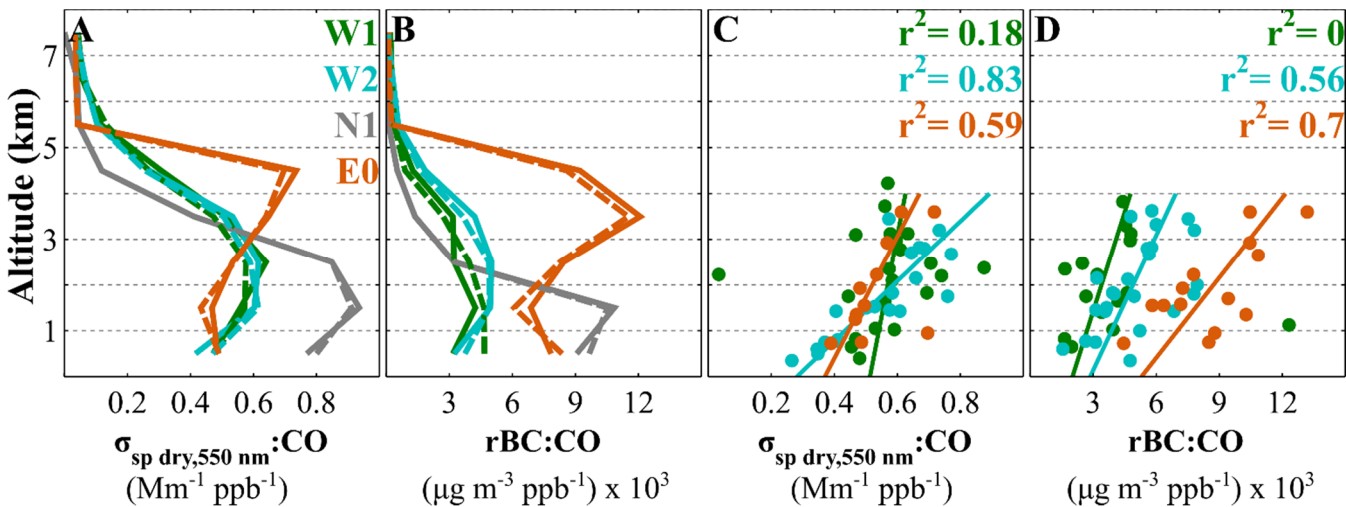

**Figure 9.** Median enhancement profiles of $\sigma_{sp\_dry}$:CO (A) and rBC:CO (B) for each regime. Solid (dashed) line represents averages with (without) plumes included. The enhancements were calculated for each individual profile by subtracting the regime 5th percentile. . Plume enhancement ratios of $\sigma_{sp\_dry}$:CO (C)and rBC:CO (D) calculated from plume integrated values above the local background (moving 5th percentile, i.e. dashed grey baseline Fig. 3) only when the two pollutants are well correlated (Pearson's r > 0.5). Ratios reported for pollutants are at ambient temperature and pressure.

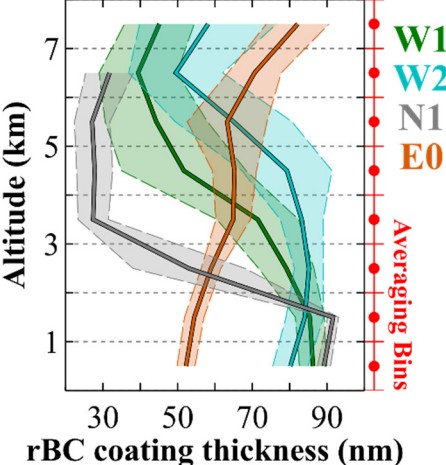

**Figure 10.** Regime median profiles of black carbon coating thickness. Lighter shading represents the median absolute deviation.

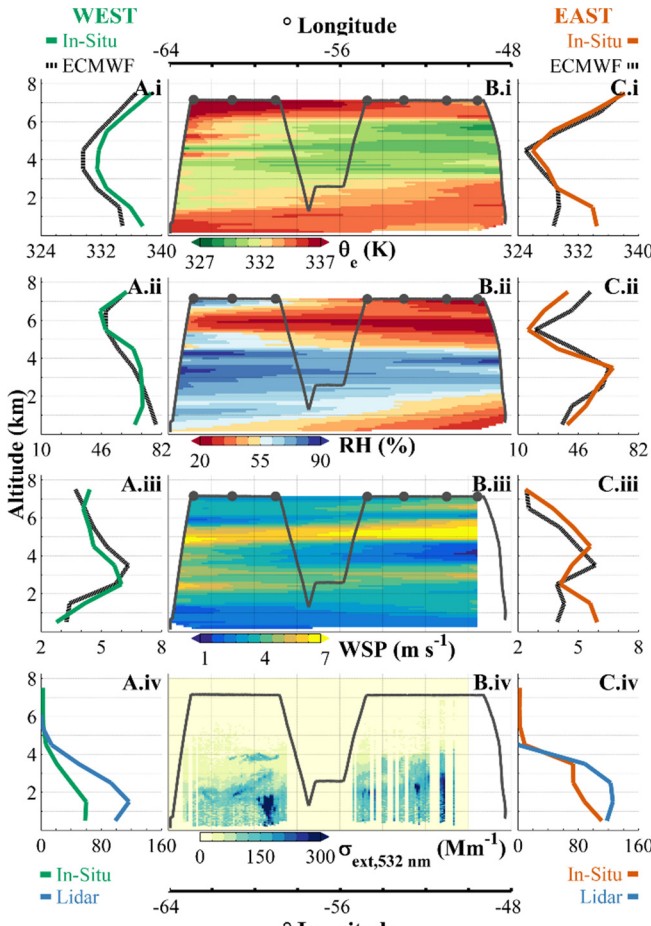

**Figure 11. Summary of west to east regional gradient in equivalent potential temperature (i), relative humidity (ii), horizontal wind speed (iii) and aerosol extinction (iv) based on an east to west transect (Panels B) on the 27th September 2012 (flight b743). The thermodynamic curtains (B.i-iii) are derived from linearly interpolated aircraft (grey line) and dropsonde (grey dots) profiles. The curtain of the lidar extinction coefficient at 532 nm (B.iv) is reproduced from Marenco et al., (2016) scaled from 355nm using a scaling factor of 0.57 following Marenco et al., (2014). The regional gradient is emphasised by the side panels which show the median thermodynamic and aerosol extinction profiles for all western (A) and eastern flights (C). In panels (i) to (ii) the observed thermodynamic parameters are compared to ECMWF reanalysis data extracted along the flight paths. In panel (iv) the in-situ aerosol scattering coefficient at 550 nm is compared to the lidar extinction coefficient averaged over a similar region (Fig. S6; n.b. both measurements are reported at ambient temperature and pressure).**