# Peer review of "The vertical distribution of biomass burning pollution over tropical South America from aircraft in situ measurements during SAMBBA"

_Atmospheric Chemistry and Physics, 2018_

## Referee Comment (RC1) · Anonymous Referee #1 · 24 Oct 2018

Reviewer's Comments on 'The vertical distribution of biomass burning pollution over tropical South America from aircraft in situ measurements during SAMBBA' by Darbyshire, et al.

General Comments

This manuscript reports findings from the 2012 SAMBBA field campaign with respect to atmospheric distributions of several key pyrogenic pollutants (smoke aerosol, black carbon, and CO). Going further, the manuscript interprets the effects of meteorology on the observed distributions. Much of the detailed information with respect to the data and methods is provided in a Supplementary file. Overall, the paper is scientifically

sound and well written, and should be interesting to many readers of ACP. Most of the suggested revisions listed below are relatively minor.

The only significant objection I have to the content is the lack of any consideration of the effects of deep convection on the transport of pollutants in the Amazon Basin. While this mechanism may become more significant towards the dry/wet transition season, a number of studies have reported finding elevated CO concentrations in the Amazonian upper troposphere, likely caused by deep convection. Among these are Andreae et al., 2001 (Geophys. Res. Lett., 28(6), 951–954, https://doi.org/10.1029/2000GL012391), Livesey et al., 2013 (Atmos. Chem. Phys., 13, 579–598, https://doi.org/10.5194/acp-13-579-2013), and Deeter et al., 2018 (J. Geophys. Res., 123, https://doi.org/10.1029/2018JD028425).

In fact, a significant number of the CO profiles presented in the author-provided Supplementary file exhibit features consistent with vertical transport via deep convection. For example, roughly between a third and a half of the 'complete' SAMBBA CO profiles (including CO measurements in the lower and upper troposphere) indicate CO enhancements in the upper troposphere (e.g., above 500 hPa) of 50 ppbv or more (relative to the minimum in the vertical profile). Interestingly, these enhancements often (but not always) appear without corresponding features in the aerosol extinction profile, possibly indicating 'rainout' of the aerosol. This feature of the CO profiles should be investigated in the revised manuscript and the potential role of deep convection should be addressed generally.

Minor Revisions and Technical Corrections

page 5, line 17. Please provide reference for climatological winds over South America (e.g., Campetella, C. M., and Vera, C. S., Geophys. Res. Lett., 29, 1826. https://doi.org/10.1029/2002GL015451, 2002).

p. 6, l. 26. Unclear if 80% refers to number of profiles where any or all of the considered pollutants indicated a pollutant residual layer.

p. 7, l. 24. What about the east-west distribution of fires? The figure indicates many more fires in the eastern region. Is this typical?

p. 8, l. 1. Might stronger easterlies actually promote (rather than inhibit) the spread of fires?

p. 8, l. 17. Add 'significantly' before 'affected'. There must be some small effect, correct?

p. 8, l. 27. It seems surprising that CO at the surface decreases from W1 to W2 (from 340 to 220 ppbv) whereas CO2 increases slightly (from 394 to 397 ppm). Does this suggest biogenic influence?

p. 9, l. 18. If this statement ('Significantly, the shift ...') is based on Fig. 9, should the end of the sentence actually read ' ... the relative abundances of rBC and sigma_sp_dry *to* CO'?

p. 9, l. 29. This particular paragraph ('The shapes of pollutant vertical distributions ...') seems largely qualitative and speculative. For example, sentences 5, 6, 7 and 9 in this paragraph draw conclusions without providing any quantitative evidence.

p. 10, l. 28. Missing 'and' between 'phase' and 'plume'?

p. 12, l. 14. SAMBBA was conducted in a year which was not considered a 'drought year' for the Amazon Basin. Widespread drought, such as occurred in 2010 and 2015, and may be increasing in frequency, results in different patterns of emissions (and meteorology) compared to non-drought years. Would the main findings of this paper be sensitive to the effects of drought? This would be an appropriate discussion for the Conclusion.

---

## Referee Comment (RC2) · Eoghan Darbyshire et al. · 5 Nov 2018

Review of Darbyshire et al smoke vertical profiles in Brazil.

The authors present a vast body of very important data on the state of the atmosphere over Amazonia as evidenced in part by an incredible 168 page supplement with about 200 vertical profiles; each with many variables. It's been six years since the data was collected indicating that time for a very significant analysis effort occurred. There is a lot to think about. Caveats on my review: I'm not a meteorology expert and I only had time to read quickly thru the text and figures twice with only a glance at the supplement.

One point to raise is that there are two different important questions: 1) what is the state of the atmosphere? and 2) what causes it? The paper seems stronger on the

first point, and information in regard to the first question is important even if the second one cannot be answered definitively.

My overall recommendation is to check the units (for BC especially) and figure captions carefully and to make sure a few angles of analysis (which I will list next) have been considered adequately. Most or all of the ideas on my list were probably considered by the authors, but since little or no mention is made, the reader can't be sure what has and has not been considered. Regarding the issues on my list, my expectation is that the authors can likely easily dismiss some, others may be sources of uncertainty in the pollutant profiles or driving forces, but yet others may yield insights. Most important is how these issues could impact the certainty about the actual pollutant structure.

Issues to address include:

1) Possible changes in the mass scattering coefficient with age (Akagi et al., 2012 in ACP).

2) The possibility of secondary organic or inorganic aerosol formation with aging.

3) The potential effects of aerosol concentration and ambient temperature on gas-particle partitioning.

Clues to these issues might possibly come from e.g., the coating thickness to BC ratio, neph-scattering to BC ratio, neph-scatter:SP2-scatter?, and O:C ratio of any AMS spectra collected or SPMS data that may have been even occasionally collected.

4) The SP2 is good for BC. Was there any info on BrC?

5) Could dust or PBAP, e.g. super-micron aerosol have a regional role that should be put in perspective?

6) Was any use made of GOES data?

7) Briefly, how was "aerosol:CO" calculated and what are it's units? Is aerosol:CO related to coating thickness and are there any insights at all from the AMS that was on

board?

Deep convection was mentioned by other reviewer and Yokelson et al., (2013, AMT) have a paper on the possibility of mixing (especially BL/FT) to change source ratios.

So in general, check all figures and units carefully and maybe do one more brainstorming run on the interpretation? A careful empirical summary of the observations without complete proof of why they occur is also useful.

Comments by line number:

P1, 16: Reduced uncertainty in processes may be less important than just presenting the wealth of vertical profile info.

P1, L22: First place I noticed where units on BC may be off?

P1, L23: Can you really measure surface CO from a plane? Here and elsewhere, maybe "surface" should be defined as an AGL range?

P1, L26: "Optically thin" or "thin layers of pollutants"?

P1, L31-33: I'm not sure about this hypothesis as explained in detailed comments. Also MCE can be hard to measure away from the source (Yokelson et al 2013 AMT), especially in the free troposphere. BC may be a good independent indicator of "flaming induced plume rise" if/when that happens.

P2, L11: Clarify what is meant by haze lifetime? Average lifetime of a smoke plume from a typical contributing fire?

P3, L2: At least four papers, including two from the FAAM group by Hodgson et al and Reddington et al seem to prove that missing fires is the biggest reason for emissions errors. The two non-FAAM papers are here:

https://agupubs.onlinelibrary.wiley.com/doi/epdf/10.1029/2018GL078679

https://www.atmos-chem-phys.net/11/6787/2011/acp-11-6787-2011.html (See Table 1

on fire detection).

P4, L9: No mention is made of the AMS. How is "aerosol" measured an what are its units?

P4, L20: Which profiles, from the plane or the dropsondes or both?

P5, L6 and L28: Is this a local maximum in wind speed not counting the free troposphere?

P5, L10 and L27: Does "positive" wind shear mean an increase in headwind experienced by the aircraft, a decrease or increase in prevailing winds, a wind direction change, or something else?

P6, L18-21: It makes sense that the atmosphere is more often well-mixed in afternoon and we and others have seen that, but why would BC be well mixed but not CO if they are from the same source?

P6, L24: delete "c" in "cm-3"?

P7, 4-5: Why are all pollutants not elevated in plumes? Variability, mixing, and other sources are mentioned and make sense especially for CO2. But for CO, BC, and scattering; maybe the threshold is too high or plume signatures are erased by mixing?

P7, L8&10: "lifted" > "lifting"

P7, L27: "Whist" > "Whilst"

P7, L30: AOD seems to peak downwind of the area of maximum fires, which is "OK". This sentence may be better as "an enhancement in AOD of 0.1," but this seems to be a very small AOD enhancement compared to the AOD of 4.0 mentioned earlier

P7, L29-32: Most of the fires are set by farmers, ranchers, or loggers, and they like to burn at low RH and wind speed to get good fuel consumption, but with reasonable ease of control/less chance of escape. An increase in either RH or wind speed means

less suitable conditions for initiating burning. However, also, increased cloud cover can reduce fire detection efficiency.

P8, L11: Does E0 cover Sep 14 – Oct 5?

P8, L18: How is "absolute aerosol burden" defined and quantified?

P8, L20-22: Again here and throughout: Am I understanding the units right? 120 Mm-1 would be about 40 ug/m3 of PM2.5 – but the rBC of 1.7 ug/cm3 would be about 1.7 E6 ug/m3. In Fig S3, the rBC peaks about 1.5 ng/sm3, which is 9 orders of magnitude lower? Then in the 198 profiles the BC is about 1 ug/m3, which is different again, but the ug/m3 units seems to make the most sense throughout.

P8, L23: "surface" should be more precisely defined since we observed some instances of very high, reasonably widespread CO (~5ppm) on the ground in the Amazon.

P8, L24: "unknown" or is e.g. "variable" more appropriate?

P8, L26-28: Is 397 ppm the surface layer CO2 in W1 and W2? Andreae et al 1988 found CO2 at low altitudes really dependent on time of day (PS vs respiration).

P8, L31: How is the "aerosol burden" calculated? E.g. is there an inferred mass? The aerosol emissions are usually higher at lower MCE in most studies, unless the authors have a good example in mind of the contrary? Seeing PM/CO (g/g/ or scat/ppm) larger at higher altitude could indicate PM evolution or that high MCE does not cause high plume rise? Other factors that influence plume rise, such as fuel consumption rate or atmospheric physics, might be more important?

P9, L2: Again excess aerosol should have units or some more precise explanation?

P9, L8-10: Here the authors do note a possible role of temperature and aging on coating thickness, which is great. Perhaps temperature and smoke age deserve broader attention?

P9, L11: Why the difference in coating thickness above the BL for different regimes? Do these regimes have a different concentration gradient above the BL so that evaporation due to a concentration decrease happens only in some cases?

P9, L19: "influence" may be better than "determine"?

P9, L28: "combustion processes" may be better than "emissions"?

P10, L1: sentence may need one more word? E.g. "they" before "contribute"?

P10, L2-3: is the relative frequency of deep convection different from Andreae et al 2004 maybe because different geographic regions were sampled?

P10, L6-7: Saying pollution persisted "above" the mixed layer seems potentially inconsistent with the earlier text about clean conditions at altitude.

P10, L9: "they exhibit"?

P10, L13: "eats" should be "east"? and L13-14: "above the lifting"?

P10, L22: In reality, it should be remembered that "plume injection altitude" is often not a fixed number, but an evolving mass-flux-vs-altitude, surface to plume top, distribution function (see Stocks et al (1996, JGR) SAFARI-92 paper).

P10, L25-34: A number of things besides MCE could influence "aerosol"/CO. For example, secondary organic or inorganic aerosol, temperature, the increase in MSC with aging, and fire intensity as noted in Reid et al., (1998) where more intense fires had much higher EF-PM similar to the finding for wild vs prescribed fires in Liu et al. (2017). Fires tend to be more intense later in the day when the mixing layer is also deeper. It's also possible the authors have the relation between MCE and EFPM reversed as noted above. I don't recall any of the cited works supporting the author's claim that aerosol/CO increases with MCE, but if they do, it might help to point out the specific figure or table where that evidence occurs? The phrase "combustion completeness phase plume injection height" is awkward and at a minimum, perhaps missing a word

or dash to be clear? Further, "combustion completeness" is often used to describe the fraction of fuel consumed and Akagi et al (2011) actually argue that "phases" do not exist on real fires. The speculation is "OK" but it might make more sense to include a few candidate ideas about what drives the "aerosol"/CO rather than try to choose one based on the limited available evidence and resulting uncertainties.

P11, L1: Maybe "remote sensing" is too broad since MOPITT a-priori profiles, and perhaps others, are flat over the whole column?

P11. L8: replace "This study" with explicit reference to a paper?

P11, L14-28: Good points.

P11, L29-33: I think the description of fires is potentially misleading and other driving factors for aerosol/CO noted above deserve mentioning here. Small fires in the tropics in homogeneous fuels often are mostly a steady mix of emissions entrained into a column by flaming. Residual smoldering could play a role, but explicitly accounting for post-convection, residual-smoldering emissions would require separate EF and separate fuel consumption, which is not trivial to estimate. Estimates are available in JGR papers by Bertschi et al 2003 and Christian et al 2007, specifically for Brazil. I think the key here is to characterize the structure and the uncertainty in the structure, with a list of possible causes (examples above) being nice, but perhaps presented without choosing a preferred option, unless additional evidence can be presented also.

P12, L3: The Reddington et al scaling factor mentioned earlier in this paper and the other papers I noted above in this review overwhelmingly suggest that missing fires is the problem rather than, e.g., a factor of five on emission factors.

P12, L1-12: I kind of glazed over trying to follow this, so I'll just interject a detailed caution against assuming MCE and plume rise are positively correlated while freely admitting that they could be related at times.

Converting fuel carbon to $CO_2$ releases more energy than converting it to CO, but

buoyancy is related to air temperature and the rate of fuel consumption or CO2 production. A fire burning with lower MCE could consume fuel at a higher rate in heavier fuels. Even for a fixed fuel loading, imagine a flame front at the beginning of a fire burning with an MCE near 1. As soon as it moves there is smoldering in its wake and MCE drops, but total CO2 is increasing for a relevant-sized area because more fuel is burning. Air temperature above the fire peaks when CO2 peaks, not MCE. That is what has been seen in most of our lab experiments or field work with towers or low flying helicopters. E.g. in Akagi et al 2012 an 81 ha fire had similar MCE at different heights in the convection column and the plume rise was dependent more or less on the instantaneous area burning. If you watch tropical fires, plume rise often peaks when flame fronts started on opposing edges converge, and then they annihilate.

Secondly, even if the plume has more thermal energy, it may not rise higher ultimately. Plume rise depends on atmospheric stability, entrainment, the vertical profile of the winds, temperature, moisture (condensation can release energy), etc. The authors are in a good position to advise on the typical thermodynamics!

Thirdly, flaming vs smoldering may impact the plume rise, but to check further maybe try some other F/S indicators like BC/CO that might work better or independently.

P12, L29-31: Christian et al 2003 (JGR), Akagi et al., (2011), Andreae & Merlet 2001, and many other papers do already recommend higher BC/OC, BC/CO, etc for savannah vs forest.

P13, L10: Is this obvious from satellites or the average meteorology?

Figure 1. Its hard to tell the difference between high altitude surveys and straight level runs, make one orange or yellow or green?

Fig 2. Should there be typical altitude values (and units) on the vertical axis?

Fig 3. There are two BB plumes identified on the rBC trace. There are no BB plumes identified on the CO trace, and only one each on CO2 and Bscat. It seems like there

should be two on all traces if they really are BB. Also, I find it hard to locate/digest the structure and plume features that are explained in the upper right boxes.

Fig 4. How can the difference for soil moisture be in tens when the absolute values are fractional?

Fig 5. Maybe label FC as "FC/100" also in Fig 6? Interesting that the AOD max is downwind of the detected fire max. This could be transport, but also different detection efficiency, higher PM emissions from more smoldering in forest fires. AODs up to 4 that are mentioned in the text relate to this figure how?

Fig 6. Would dates or definitions of dry and dry-wet transition seasons be useful? Yokelson et al., (2007) saw a huge burst in burning during a few days with the lowest RH.

Fig 7. The figure doesn't appear to have any "pollutant" info that is referred to in the caption. Do you mean "meteorological"?

Fig 8. Check units on rBC. Could the CO and aerosol shapes be different partly because CO may be more impacted by non-fire (urban) sources that inject lower than fires?

Fig 9. Needs units. More?

Fig 10. Any way to relate coating thickness to age, or size, or Bscat/BC or T, where the latter two might be related to gas to particle partitioning?

Fig 11. This is kind of overwhelming. Maybe help the reader with a few sentences about what it is trying to show.

---

## Author Comment (AC1) · 11 Mar 2019

We thank the anonymous reviewer for her/his thorough evaluation and constructive recommendations for improving this manuscript. Her/his comments (in italics) and our responses are listed below.

*This manuscript reports findings from the 2012 SAMBBA field campaign with respect to atmospheric distributions of several key pyrogenic pollutants (smoke aerosol, black carbon, and CO). Going further, the manuscript interprets the effects of meteorology on the observed distributions. Much of the detailed information with respect to the data and methods is provided in a Supplementary file. Overall, the paper is scientifically sound and well written, and should be interesting to many readers of ACP. Most of the suggested revisions listed below are relatively minor.*

*The only significant objection I have to the content is the lack of any consideration of the effects of deep convection on the transport of pollutants in the Amazon Basin. While this mechanism may become more significant towards the dry/wet transition season, a number of studies have reported finding elevated CO concentrations in the Amazonian upper troposphere, likely caused by deep convection. Among these are Andreae et al., 2001 (Geophys. Res. Lett., 28(6), 951–954, https://doi.org/10.1029/2000GL012391), Livesey et al., 2013 (Atmos. Chem. Phys., 13, 579–598, https://doi.org/10.5194/acp-13-579-2013), and Deeter et al., 2018 (J. Geophys. Res., 123, https://doi.org/10.1029/2018JD028425).*

*In fact, a significant number of the CO profiles presented in the author-provided Supplementary file exhibit features consistent with vertical transport via deep convection. For example, roughly between a third and a half of the 'complete' SAMBBA CO profiles (including CO measurements in the lower and upper troposphere) indicate CO enhancements in the upper troposphere (e.g., above 500 hPa) of 50 ppbv or more (relative to the minimum in the vertical profile). Interestingly, these enhancements often (but not always) appear without corresponding features in the aerosol extinction profile, possibly indicating 'rainout' of the aerosol. This feature of the CO profiles should be investigated in the revised manuscript and the potential role of deep convection should be addressed generally.*

This is an important point and one mostly missed in the discussion manuscript. Based on the recommendations an analysis was conducted on each individual profile that extended through sufficient depth of the troposphere to identify enhancements in CO linked to deep convection. CO enhancements above the boundary layer were observed in 80% of profiles. However, only rarely was a similar increase seen in the aerosol products, indicating rainout as the referee suggests. This is consistent with the rBC coating thickness measurements. A summary of the changes to the supplement and manuscript to reflect this new topic area is provided below:

Method description: Sect S2, Supplement
Pollutant transport via deep convection: CO profiles with a sufficient vertical extent, of at least 5 km, were identified. The altitude of the CO minimum was identified and was used to determine a representative background concentration for the altitudes above and the altitude below. The altitude of the CO maximum above the minimum was identified and a representative CO maximum calculated from the concentration at that altitude, the altitude above and the altitude below. If the representative maximum CO concentration was greater than the representative minimum CO concentration plus a threshold of 40 ppb, transport via deep convection was identified. The rBC and $\sigma sp$ values were interrogated at the CO minimum and the altitudes of the maxima. If these were greater at the maximum altitude by 0.2 µg m-3 (rBC) or 25 Mm-1 ($\sigma sp$) than transport via deep convection for these aerosol properties was also identified.

Results: Table S3, Supplement

|  | All | | E0 | | W1 | | W2 | | N1 | |
|---|---|---|---|---|---|---|---|---|---|---|
| CO Deep convection | 81.1 | (53) | 100 | (2) | 77.8 | (18) | 79.3 | (29) | 100 | (4) |
| Co-incident rBC increase | 8.1 | (37) | 0 | (2) | 10.0 | (10) | 9.5 | (21) | 0.0 | (4) |

| Co-incident $\sigma_{sp}$ increase | 2.7 (37) | 0 (2) | 0.0 (11) | 4.8 (21) | 0.0 (3) |

Table S3. Percentage of sufficiently deep profiles showing evidence of CO transport via deep convection and co-incident increase in rBC and $\sigma_{sp}$. Bracketed values represent the number of sufficiently deep profiles.

Manuscript results, Sect 3.2:
There is evidence of moist convection delivering CO to altitudes above ~4 km but with significant wet scavenging of aerosol. In 81% of profiles with sufficient vertical coverage CO loadings increased by more than 40 ppb at altitude between ~4km to the top of the profile. Unlike the discrete signal from plumes, the enhancement was often 1-2 km deep. In contrast only 8% and 3% of the rBC and $\sigma_{sp}$ profiles had a similar increase in signal co-incident with CO enhancements. This indicates significant removal of aerosol during convection to altitudes above ~4km across the atmosphere of Amazonas during the biomass burning season.

Manuscript discussion, Sect 4.1:
The shapes of pollutant vertical distributions are primarily controlled by meteorological conditions, in particular vertical convective motions and horizontal wind shear (Fig. 7). The former acts to mix pollutants released near the surface toward the mixing layer top, the altitude of which can be modulated by the latter, soil moisture and solar insolation. The difference in profile shape from west to east to north is primarily driven by contrasting mixed layer depths. Pollutant loadings remained relatively high above the mixing layer in residual layers, indicating wet removal is not significant at these altitudes. Large unmixed plumes perturbed the mixed and residual layers, although they contribute only 15% (E0), 11% (W1), 8% (W2), and 1% (N1) to the scattering only column AOD (calculated from the nephelometer, Sect. S2). Such plumes were seldom seen above 4 km, in contrast to previous observations in the region (Andreae et al., 2004), indicating the mass flux from large pyrocumulus detrainment into the upper troposphere (within the aircraft range) was not significant. The observed increases in CO concentrations above ~4km indicate vertical transport of mixing layer pollution into the free troposphere. The presence of co-incident increases in rBC or $\sigma_{sp\_dry}$ in less than 10% of these plumes indicates moist deep convection transports CO and presumably other gaseous and non-soluble components to altitudes greater than 4 km but efficiently removes aerosol from the air by wet scavenging. This is consistent with the decrease in rBC coating thickness at these altitudes in W1, W2 and N1 and also is similar to previous observations in Boreal Canada that showed preferential wet deposition of the largest and most coated particles (Taylor et al., 2014). As deep moist convection is not common in eastern regions (e.g. using TRMM rainfall as a proxy, Fig. 4a) the source of elevated and enhanced CO is unlikely to arise from the mixing layer in the east. It is possible the source is from CO aloft in the west which is recirculated in the persistent anti-cyclonic flow at 500 hPa (Fig. S2.f) as has previously been observed from satellite (MOPPIT) CO observations by (Deeter et al., 2018). A long aging time is consistent with the larger rBC coatings observed aloft in E0.

**Minor Revisions and Technical Corrections**
page 5, line 17. Please provide reference for climatological winds over South America (e.g., Campetella, C. M., and Vera, C. S., Geophys. Res. Lett., 29, 1826. https://doi.org/10.1029/2002GL015451, 2002).
Thank you for the reference, this has now been included:
'In general, observed profiles of horizontal wind speed reflect those expected based on our understanding of synoptic flows over TSA (Campetella and Vera, 2002).'

p. 6, l. 26. Unclear if 80% refers to number of profiles where any or all of the considered pollutants indicated a pollutant residual layer
The sentence has been amended to make it clearer and now includes a cross-reference back to the originating table.
'Over 70% of profiles included a pollutant residual layer of rBC, $\sigma_{sp}$ and CO, even those in remote regions away from fresh emissions (Table S1).'

p. 7, l. 24. What about the east-west distribution of fires? The figure indicates many more fires in the eastern region. Is this typical?

The distribution is typical in that fire number is greatest in the east, although 2012 did feature more fires in the east than the 2008-2017 average. We decided against including a detailed climatology of fires and trends in the paper to keep it to a more readable length and focus the narrative.

p. 8, l. 1. Might stronger easterlies actually promote (rather than inhibit) the spread of fires?

If we were dealing with wildfires that may be true – however as most of the fires are managed by landowners, they tend to ignite when wind speed is low to enable greater control of the burn. This section has been rewritten based on the more detailed comments by reviewer 2.

p. 8, l. 17. Add 'significantly' before 'affected'. There must be some small effect, correct?

This has been amended – there is a small effect rather than no effect.

p. 8, l. 27. It seems surprising that CO at the surface decreases from W1 to W2 (from 340 to 220 ppbv) whereas CO2 increases slightly (from 394 to 397 ppm). Does this suggest biogenic influence?

Yes, that is correct, this was noted on P9 L5 of the original manuscript. P9 L18 in updated manuscript.

p. 9, l. 18. If this statement ('Significantly, the shift …') is based on Fig. 9, should the end of the sentence actually read ' … the relative abundances of rBC and sigma_sp_dry *to* CO'?

Thanks to the referee for pointing this out.  The sentence has been modified to read:

Significantly, the shift in meteorology between these two phases does not substantially impact the relative abundances of rBC, $\sigma$sp_dry and CO to each other.

p. 9, l. 29. This particular paragraph ('The shapes of pollutant vertical distributions …') seems largely qualitative and speculative. For example, sentences 5, 6, 7 and 9 in this paragraph draw conclusions without providing any quantitative evidence.

This has been revised (see the text from section 4.1) in the response to the general comment above.

p. 10, l. 28. Missing 'and' between 'phase' and 'plume'?

This section has been re-written following comments by reviewer 2.

p. 12, l. 14. SAMBBA was conducted in a year which was not considered a 'drought year' for the Amazon Basin. Widespread drought, such as occurred in 2010 and 2015, and may be increasing in frequency, results in different patterns of emissions (and meteorology) compared to non-drought years. Would the main findings of this paper be sensitive to the effects of drought? This would be an appropriate discussion for the Conclusion.

We thank the reviewer for a useful suggestion to widen the discussion. The following paragraph has been included in Discussion-Implications (Sect 4.2):

'Although the magnitude and bearing may differ, the fundamental drivers of the pollutant vertical distribution identified here will remain so in drought years which may be increasing in frequency (Jiménez-Muñoz et al., 2016). Dry convection may be more vigorous and the atmosphere more stable, deep convection less vigorous and aerosol scavenging reduced, fires more intense and fire hotspots located in different regions, but so long as model simulations well represent the fundamental drivers identified in this work then they ought to be able to replicate the resultant vertical distribution. This is a promising avenue for future research to predict the impacts in future years, following on from the study of Thornhill et al., (2018).'

---

## Author Comment (AC2) · 11 Mar 2019

We thank the anonymous reviewer for her/his thorough evaluation and constructive recommendations for improving this manuscript. Her/his comments (in blue) and our responses are listed below.

The authors present a vast body of very important data on the state of the atmosphere over Amazonia as evidenced in part by an incredible 168 page supplement with about 200 vertical profiles; each with many variables. It's been six years since the data was collected indicating that time for a very significant analysis effort occurred. There is a lot to think about. Caveats on my review: I'm not a meteorology expert and I only had time to read quickly thru the text and figures twice with only a glance at the supplement.

One point to raise is that there are two different important questions: 1) what is the state of the atmosphere? and 2) what causes it? The paper seems stronger on the first point, and information in regard to the first question is important even if the second one cannot be answered definitively.

My overall recommendation is to check the units (for BC especially) and figure captions carefully and to make sure a few angles of analysis (which I will list next) have been considered adequately. Most or all of the ideas on my list were probably considered by the authors, but since little or no mention is made, the reader can't be sure what has and has not been considered. Regarding the issues on my list, my expectation is that the authors can likely easily dismiss some, others may be sources of uncertainty in the pollutant profiles or driving forces, but yet others may yield insights. Most important is how these issues could impact the certainty about the actual pollutant structure.

Issues to address include:
1) Possible changes in the mass scattering coefficient with age (Akagi et al., 2012 in ACP).
See the comment below

2) The possibility of secondary organic or inorganic aerosol formation with aging.
The above two points relate to changes in aerosol composition and therefore optical properties with age. This is the focus of a recent paper by Morgan et al., (2019), which investigates aging of biomass burning aerosol in both the near and far field based on SAMBBA measurements. We note that there is no indication in this work of substantial increases in the mass scattering coefficient with age. Some near field changes in the mass scattering coefficient with age have previously been observed (Akagi et al., 2012). However, the work here is focussing on regional distributions and changes and individuals fire plumes are not a significant component of the study and are unlikely to affect the results presented here.

3) The potential effects of aerosol concentration and ambient temperature on gas particle partitioning. Clues to these issues might possibly come from e.g., the coating thickness to BC ratio, neph-scattering to BC ratio, neph-scatter:SP2-scatter?, and O:C ratio of any AMS spectra collected or SPMS data that may have been even occasionally collected. These issues are addressed as part of a new section, which covers many of the line by line comments regarding the source of aerosol aloft. See (a.) below.

4) The SP2 is good for BC. Was there any info on BrC?
Unfortunately not – no multi wavelength absorption measurements were made using the aircraft.

5) Could dust or PBAP, e.g. super-micron aerosol have a regional role that should be put in perspective?
We examined the aerosol size distributions from the SMPS and GRIMM and also determined the Angstrom exponent at Red/Green to show that there was little contribution of dust or biological particles to the optical properties of aerosol in the column. We address this in the manuscript with the following figure and sentence in section 4.1:

'Consistent values of the scattering angstrom exponent at 700/550 nm within the boundary layer (Fig. S5) indicate a similar aerosol type throughout. These values were all above 1.5, typical of submicron biomass burning aerosol and indicating no significant regional role for super-micron dust or primary biological aerosols which have values closer to zero (Clarke et al., 2007; Russell et al., 2010). This is consistent with size distributions reported and from the aircraft (Darbyshire et al., in preparation).'

[Figure]

Fig S5. Median profiles of the scattering Angstrom exponent between 550 and 700 nm for each regime. Angstrom exponent calculated from nephelometer scattering coefficient measurements at these wavelengths. To reduce the noise the Angstrom exponent is only displayed here when the scattering coefficient at 550nm was greater than 10 $Mm^{-1}$. Lighter shading represents the median absolute deviation.

6) Was any use made of GOES data?

The resolution provided by MODIS data provides better spatial coverage than GOES and additional discussion of further satellite information was felt unnecessary in what is already a lengthy paper.

7) Briefly, how was "aerosol:CO" calculated and what are it's units? Is aerosol:CO related to coating thickness and are there any insights at all from the AMS that was on board?

Aerosol:CO is used as a generic term to represent both scattering:CO and rBC:CO ratios which displayed similar behaviour. These enhancement ratios were calculated from plume integrated values above the local background (moving 5th percentile, i.e. grey 'background' in Fig. 3) and only when the two pollutants were correlated (Pearson's r > 0.5). We are sorry that this was confusing and have added a sentence in section 3.4 of revised text to make this clear:

'We use the term $\Delta$aerosol:$\Delta$CO to describe both $\Delta$rBC:$\Delta$CO and $\Delta$ $\sigma_{sp\_dry}$:$\Delta$CO since these aerosol properties co-vary.'.

There was an AMS on-board, the vertical distributions from which will be presented in a forthcoming publication from this author, which presents detailed information on the aerosol composition and physical properties

Deep convection was mentioned by other reviewer and Yokelson et al., (2013, AMT) have a paper on the possibility of mixing (especially BL/FT) to change source ratios.

We thank the reviewer for the reference and note the point with the following added to section 3.4:

'As these enhancement ratios are typically within the boundary layer, share a common source and in most cases are likely relatively fresh, it is unlikely they are biased by sudden changes in the composition of background air driving the observed gradient, as warned against in Yokelson et al. (2013).'

We reply to the issue of deep convection in response #1.

So in general, check all figures and units carefully and maybe do one more brainstorming run on the interpretation? A careful empirical summary of the observations without complete proof of why they occur is also useful.

We have now checked all the figures thoroughly.

Comments by line number:

Where the referee has made a number of comments on one subject, we have grouped the comments together and provide a combined response:

a) Cause of increasing aerosol:CO with altitude

We thank the reviewer for their detailed comments on this section, which have led to a significant re-working that has, we believe, improved the paper.  The following comments are addressed in a re-written section which can be found below. Where comments are not directly addressed in this new text we have responded underneath that comment.

P1, L31-33:  I'm not sure about this hypothesis as explained in detailed comments. Also MCE can be hard to measure away from the source (Yokelson et al 2013 AMT), especially in the free troposphere. BC may be a good independent indicator of "flaming induced plume rise" if/when that happens.
See below

P6, L18-21: It makes sense that the atmosphere is more often well-mixed in afternoon and we and others have seen that, but why would BC be well mixed but not CO if they are from the same source?
See below

P7, 4-5: Why are all pollutants not elevated in plumes? Variability, mixing, and other sources are mentioned and make sense especially for CO2. But for CO, BC, and scattering; maybe the threshold is too high or plume signatures are erased by mixing?
This is the crux of what we are trying to understand, not only are the ratios of the aerosol properties to CO elevated in the averaged profiles, but also within the plumes. Mixing ought to mix out all the species equally, not some but not others, provided that their sources are all the same. We do not believe that we alias the ratios in the plumes as a result of the plume thresholds being set too high.  We do show all of the profiles in the supplementary and there are no indications of enhanced values of variables sitting below the thresholds co-incident with plumes of other variables that exceed them.  Unfortunately, because of the variability in the CO2 concentration, mentioned by the reviewer, it was not possible to reliably retrieve MCE as a function of height which would have helped strengthen our hypothesis.

P8,L31: The aerosol emissions are usually higher at lower MCE in most studies, unless the authors have a good example in mind of the contrary?
The referee is correct that aerosol emissions do increase as MCE decreases and we have certainly observed this previously in laboratory and single plume studies.  However, since we are comparing aerosol emissions to CO it may be that even though the aerosol emissions may increase as MCE decreases the proportional increase in CO is greater.

Seeing PM/CO (g/g/ or scat/ppm) larger at higher altitude could indicate PM evolution or that high MCE does not cause high plume rise? Other factors that influence plume rise, such as fuel consumption rate or atmospheric physics, might be more important?
See below

P10, L22:
In reality, it should be remembered that "plume injection altitude" is often not a fixed number, but an evolving mass-flux-vs-altitude, surface to plume top, distribution function (see Stocks et al (1996, JGR) SAFARI-92 paper).

P10, L25-34:
 A number of things besides MCE could influence "aerosol"/CO. For example, secondary organic or inorganic aerosol, temperature, the increase in MSC with aging, and fire intensity as noted in Reid et al., (1998) where more intense fires had much higher EF-PM similar to the finding for wild vs prescribed fires in Liu et al. (2017).
In the new manuscript section below we discuss how for a given burn the fire intensity (as in heat released per unit length of fire front) is likely linked to the fire phase. This is simplified by the figure below:

[Figure]

Figure. Idealised cartoon showing link between MCE and heat flux (y axis) and fire genesis (x axis) for a hypothetical single fire. (n.b. not included in manuscript)

As the reviewer points out, it is possible that this relationships fall down for very intense fires. Previous studies have shown that as an oxygen deficiency in a large flaming zone can reduce the combustion efficiency and therefore the emission of aerosol:CO would decline (Ward and Hardy, 1991). As far as we are aware there is limited study of this in the scientific literature after this study and therefore it is not possible to comment on how likely or frequent this phenomenon will be in tropical South America. We note that In the SCAR-B project referred to, Reid and Hobbs (1996, doi https://doi.org/10.1029/98JD00159) looked at how fire intensity controlled size parameters rather than EF-PM in Amazonian fires. As such, we recommend investigation into oxygen deprivation in active flame zones as an area for future research.

Fires tend to be more intense later in the day when the mixing layer is also deeper.
The referee makes a fair point. Under the assumption fires remain a similar size then these more intense fires could be characterised by a shift the upper limit of the y-axis in the above figure, meaning one would expect a greater injection height but also a higher integrated MCE. Therefore, assuming a similar fuel bed, one would expect to see the aerosol:CO increase with height as observed here. Note, that there would be a complicating factor given the (typical) strengthening of atmospheric stability in the daytime.

It's also possible the authors have the relation between MCE and EFPM reversed as noted above. I don't recall any of the cited works supporting the author's claim that aerosol/CO increases with MCE, but if they do, it might help to point out the specific figure or table where that evidence occurs?
In Figs S8 of the supplementary material, we show the EFaerosol vs MCE using data from Ferek et al., (1998) who measured PM4 and Yokelson et al., 2007 who measured PM10. In figure S9, we show values from Hodgson et al., (2018).

The phrase "combustion completeness phase plume injection height" is awkward and at a minimum, perhaps missing a word or dash to be clear? Further, "combustion completeness" is often used to describe the fraction of fuel consumed and Akagi et al (2011) actually argue that "phases" do not exist on real fires. The speculation is "OK" but it might make more sense to include a few candidate ideas about what drives the "aerosol"/CO rather than try to choose one based on the limited available evidence and resulting uncertainties.
See below

P11, L29-33: I think the description of fires is potentially misleading and other driving factors for aerosol/CO noted above deserve mentioning here. Small fires in the tropics in homogeneous fuels often are mostly a steady mix of emissions entrained into a column by flaming. Residual smoldering could play a role, but explicitly accounting for post-convection, residual-smoldering emissions would require separate EF and separate fuel consumption, which is not trivial to estimate. Estimates are available in JGR papers by Bertschi et al 2003 and Christian et al 2007, specifically for Brazil.
See below

I think the key here is to characterize the structure and the uncertainty in the structure, with a list of possible causes (examples above) being nice, but perhaps presented without choosing a preferred option, unless additional evidence can be presented also.
See revised text below

P12, L1-12: I kind of glazed over trying to follow this, so I'll just interject a detailed caution against assuming MCE and plume rise are positively correlated while freely admitting that they could be related at times. Converting fuel carbon to CO2 releases more energy than converting it to CO, but buoyancy is related to air temperature and the rate of fuel consumption or CO2 production. A fire burning with lower MCE could consume fuel at a higher rate in heavier fuels. Even for a fixed fuel loading, imagine a flame front at the beginning of a fire burning with an MCE near 1. As soon as it moves there is smoldering in its wake and MCE drops, but total CO2 is increasing for a relevant-sized area because more fuel is burning. Air temperature above the fire peaks when CO2 peaks, not MCE. That is what has been seen in most of our lab experiments or field work with towers or low flying helicopters. E.g. in Akagi et al 2012 an 81 ha fire had similar MCE at different heights in the convection column and the plume rise was dependent more or less on the instantaneous area burning. If you watch tropical fires, plume rise often peaks when flame fronts started on opposing edges converge, and then they annihilate. Secondly, even if the plume has more thermal energy, it may not rise higher ultimately. Plume rise depends on atmospheric stability, entrainment, the vertical profile of the winds, temperature, moisture (condensation can release energy), etc. The authors are in a good position to advise on the typical thermodynamics! Thirdly, flaming vs smoldering may impact the plume rise, but to check further maybe try some other F/S indicators like BC/CO that might work better or independently.
See the revised text below

[revised manuscript text omitted]

**b) Scaling Factors**

P3, L2: At least four papers, including two from the FAAM group by Hodgson et al and Reddington et al seem to prove that missing fires is the biggest reason for emissions errors. The two non-FAAM papers are here: https://agupubs.onlinelibrary.wiley.com/doi/epdf/10.1029/2018GL078679 https://www.atmos-chem-phys.net/11/6787/2011/acp-11-6787-2011.html (See Table 1 on fire detection).
P12, L3: The Reddington et al scaling factor mentioned earlier in this paper and the other papers I noted above in this review overwhelmingly suggest that missing fires is the problem rather than, e.g., a factor of five on emission factors.

For reference, here is P3, L2:
"Furthermore, to ensure consistency with remotely sensed AOD measurements, emissions are typically up-scaled by a factor of two to five (Johnson et al., 2016; Reddington et al., 2016). There is no established physical understanding of this requirement, with deficiencies in both emissions inventories and model process posited. As such, scaling factors represent a key uncertainty in model treatment of biomass burning across the globe"
And P12, L3:
"Past satellite observations have suggested residual smouldering is a large source of CO and not fully captured by emissions inventories (Deeter et al., 2016; Pechony et al., 2013), which may be based on inappropriate emissions factors or arise from small fires outside the detection limit of active fire detection schemes (Giglio et al., 2016; Schroeder et al., 2008)."

Scaling factors to reconcile model and satellite or AERONET AOD are likely required because of two components, deficiencies in emissions inventories and deficiencies in model treatment. The reviewer contends that the deficiencies in emission inventories are not caused by the emissions factors, rather a misrepresentation of burned area owing to undetected smaller fires. Such a misrepresentation is observed in Florida by Nowell et al., 2018, however it is not possible to determine if this would successfully reconcile modelled and measured AOD. This is because Florida is a humid environment and water uptake would likely increase the AOD. As the amount of water uptake is itself an uncertain process, it is not currently possible to derive a 'dry' AOD with which to compare 'dry' model output. Therefore one cannot definitively say that '*missing fires is the biggest reason for emissions errors'.* Even if this were the case for Florida, it is not clear that the discrepancy between real and observed burned area would be the higher, lower or the same in tropical South America, where burning practices differ.

However, there is evidence from SAMBBA which does point towards the problems with emissions being from missing fires - the emissions factors in Hodgson et al., (2018) are consistent with those previously reported for the region, whilst Reddington et al., 2016 and 2018 show that using an emission inventory which includes small fires (FINN1) compared to one which doesn't (GFED3) does reduce the scaling factor required to 1.5. Again it is not possible to evaluate the overall effect because of the complicating factor of humidity, as explored by Johnson et al., (2016), Reddington et al., (2018) and a forthcoming paper from this author. Therefore, it cannot be discounted that emission factors may still not best represent emissions from biomass burning in tropical South America.

This is summarised by the following amendments to the manuscript:
Update to P3, L2 Original manuscript. (P2, L34 updated manuscript)
"Furthermore, to ensure consistency with remotely sensed AOD measurements, aerosol emissions are typically up-scaled by a factor of two to five (Johnson et al., 2016; Reddington et al., 2016). This represents a key uncertainty in model treatment of biomass burning aerosol, with deficiencies in both emissions inventories and model process posited."

Sect 4.1
'Consistent values of the scattering angstrom exponent at 700/550 nm within the boundary layer (Fig. S5) indicate a similar aerosol type throughout. These values were all above 1.5, typical of submicron biomass burning aerosol and indicating no significant regional role for super-micron dust or primary biological aerosols which have values closer to zero (Clarke et al., 2007; Russell et al., 2010). This is consistent with size distributions reported and from the aircraft

(Darbyshire et al., in preparation). This also indicates that model scaling factors to match remotely sensed AOD are not significantly biased by non-biomass burning aerosol. The observed increase in humidity with altitude in the boundary layer (Fig. 7C) will likely have a significant impact on AOD and therefore the required scaling factor as it remains an uncertain model process (Johnson et al., 2016; Reddington et al., 2018). We note that findings from SAMBBA suggest that omission of burned area from small undetected fires is the most significant source of inventory under-representation of aerosol emissions (Hodgson et al., 2018; Reddington et al., 2016b), as in other burning regions (Nowell et al., 2018).'

Update to P12, L3 original manuscript. Moved to section 4.2 (P13, L17 in updated manuscript)
We speculate the Δaerosol:ΔCO profile may be driven by a coupling between combustion efficiency and plume dynamics. If confirmed by further enquiry then future modelling studies will have to consider how best to represent the phenomenon. Replicating the gradient Δaerosol:ΔCO may be particularly important for model simulations which draw results based on a realistic vertical distribution. For example aerosol-cloud and aerosol-radiation interactions, or surface air quality simulations. Emissions factors for residual smouldering combustion in the Amazon have been collected (Bertschi et al., 2003; Christian et al., 2007), and could be used to test predictions from novel (and non-trivial) model set-ups against surface and satellite observations. Past satellite observations have suggested residual smouldering is a large source of CO and not fully captured by emissions inventories (Deeter et al., 2016; Pechony et al., 2013).

**c) Units**
P1, L22: First place I noticed where units on BC may be off?
P6, L24: delete "c" in "cm-3"?
P8, L20-22: Again here and throughout: Am I understanding the units right? 120 Mm-1 would be about 40 ug/m3 of PM2.5 – but the rBC of 1.7 ug/cm3 would be about 1.7 E6 ug/m3. In Fig S3, the rBC peaks about 1.5 ng/sm3, which is 9 orders of magnitude lower? Then in the 198 profiles the BC is about 1 ug/m3, which is different again, but the ug/m3 units seems to make the most sense throughout.

The referee is correct and many thanks for identifying the error. After tracking, the mistake arose from the units previously being in $\mu g \; sm^{-3}$, where 's' inferred standard atmosphere – this was advised against in a co-author review and inexplicably the 's' was changed to a 'c' rather than removing completely. Apologies for the confusion caused.

**d) Remaining comments**
P1, 16: Reduced uncertainty in processes may be less important than just presenting the wealth of vertical profile info.
We agree that the paper itself does not reduce uncertainty of impacts on weather and climate so we have rewritten this sentence:
We examine processes driving the vertical distribution of biomass burning pollution following an integrated analysis of over 200 pollutant and meteorological profiles measured in-situ during the South American Biomass Burning Analysis (SAMBBA) field experiment. This study will aid future work examining the impact of biomass burning on weather, climate and air quality

P1, L23: Can you really measure surface CO from a plane? Here and elsewhere, maybe "surface" should be defined as an AGL range?
The text has been amended to be consistent with language elsewhere in the manuscript via use of the term 'near surface'.

P1, L26: "Optically thin" or "thin layers of pollutants"?
In most cases, the plumes above the mixing layer are both altitudinally thin and therefore when integrating the extinction coefficient to derive the AOD, also optically thin when compared to those measured in the boundary layer. However as we don't consider the extinction coefficient in this manuscript, and to avoid the confusion you have identified, we have amended the sentence to simply read 'Thin layers above the mixing layer....'.

P2, L11: Clarify what is meant by haze lifetime? Average lifetime of a smoke plume from a typical contributing fire?
P2,L11 for reference:
"These uncertainties are pronounced in tropical South America (TSA), one of the largest global biomass burning sources.  Aerosol accumulates within the convective boundary layer forming a regional haze with a  that can cover up to 6 million km2 (Prins et al., 1998) and reach aerosol optical depths (AOD) of up to 4 in the mid-visible during particularly polluted years (Artaxo et al., 2013)."

I have removed this section of the sentence to avoid confusion as the 4-5 days figure relates only to aerosol and is estimated based on the method and figure 7 in Edwards et al, (2006) and will be uncertain.

 No mention is made of the AMS. How is "aerosol" measured an what are its units?
In the linked reference (Trembath, 2013) there is an analysis of the Rosemount inlet transmission efficiency for different particle sizes. The number and mass size distribution of biomass burning aerosol is dominated by submicron aerosols, and there is high particle transmission efficiency in this region (see Trembath, 2013). The SP2 is a sub-micron instrument. The AMS is not used in this analysis.

P4, L20: Which profiles, from the plane or the dropsondes or both?
Both, the sentence has been amended to include 'Individual profiles from the aircraft and dropsondes were averaged…'

P5, L6 and L28: Is this a local maximum in wind speed not counting the free troposphere?
L6, refers to the trade wind inversion marking the border between the PBL and free troposphere. L28 refers to our observation which show that multiple wind speed maxima were observed through the column and one of these would typically be co-located with the top of the mixing layer. This has been clarified in the text with the following sentence:

At L6
Typically, a maximum in horizontal wind speed of variable magnitude and extent is present above the entrainment zone and referred to as the trade wind inversion.

At L28
A wind speed maximum is generally observed above or collocated with the mixed layer top (e.g. Fig. 3). At times this was the trade wind inversion but typically it was not.

P5, L10 and L27: Does "positive" wind shear mean an increase in headwind experienced by the aircraft, a decrease or increase in prevailing winds, a wind direction change, or something else?
We thank the referee for pointing out that this wasn't clear in our original manuscript. This has been amended in the supplementary material (P3,L31) to include an explicit explanation:
'Positive wind shear refers to an increase in horizontal wind speed with increasing altitude.'

Thanks to the referee for pointing this out since it has also raised an omission that we did not include information regarding the wind measurements in the methodology. Hence the following sentence has been added to the methodology section.

'A five hole radome-mounted turbulence probe at the aircraft nose provides measurement of airflow relative to the aircraft, thus allowing calculation of wind vectors when combined with a GPS inertial navigation unit (Petersen and Renfrew, 2009).'

P7, L8&10: "lifted" > "lifting"
Amended

P7, L27: "Whist" > "Whilst"
Amended

P7, L30: AOD seems to peak downwind of the area of maximum fires, which is "OK". This sentence may be better as "an enhancement in AOD of 0.1," but this seems to be a very small AOD enhancement compared to the AOD of 4.0 mentioned earlier
We thank the referee for this comment. The AOD of 4 referred to earlier was an example of the extreme loadings possible as part of the introductory contextualisation. The average AOD map presented in Fig. 5a provides a more accurate representation of a typical year. The sentence has been amended based on this recommendation, from:
"Aerosol accumulates within the convective boundary layer forming a regional haze with a lifetime of 4-5 days (Edwards et al.,2006) that can cover up to 6 million km2 (Prins et al., 1998) and reach aerosol optical depths (AOD) of up to 4 in the midvisible during particularly polluted years (Artaxo et al., 2013)."
To:

"Aerosol accumulates within the convective boundary layer forming a regional haze that can cover up to 6 million km$^2$ (Prins et al., 1998) and whilst  weekly averaged aerosol optical depths (AOD) are typically 0.75-1 in the  mid visible, they can reach 4 in extremely polluted years associated with drought (Artaxo et al., 2013). "

P7, L29-32: Most of the fires are set by farmers, ranchers, or loggers, and they like to burn at low RH and wind speed to get good fuel consumption, but with reasonable ease of control/less chance of escape. An increase in either RH or wind speed means less suitable conditions for initiating burning. However, also, increased cloud cover can reduce fire detection efficiency.

The referee makes some very valuable points.  The section has been amended to:

'We speculate conditions were not optimal for human ignited fires, as a rise in relative humidity and increase in wind speed makes ignition and control more difficult, whilst decreasing the fuel consumption. The increase in cloud cover may have also reduced fire detection efficiency.'

P8, L11: Does E0 cover Sep 14 – Oct 5?

The referee makes a fair point.  It is not clear that E0 and N1 do not represent the whole period. The bullet points have been amended to account for this.

P8, L18: How is "absolute aerosol burden" defined and quantified?

Aerosol burden was the collective term employed to collectively describe both the rBC mass loading and scattering coefficient. However we acknowledge this may be confusing and misinterpreted as solely mass. As such all instances of 'burden' have been changed to 'abundance'.

The use of 'absolute' is perhaps confusing and has been removed from the sentence.

P8, L23: "surface" should be more precisely defined since we observed some instances of very high, reasonably widespread CO (_5ppm) on the ground in the Amazon. P8, L24: "unknown" or is e.g. "variable" more appropriate?

See comment for P1, L23.

P8, L26-28: Is 397 ppm the surface layer CO2 in W1 and W2? Andreae et al 1988 found CO2 at low altitudes really dependent on time of day (PS vs respiration).

We agree that this sentence is confusing – it has now been amended to:

'$CO_2$ mixing ratios are also greatest at the near surface: 402 ppm in E0, 393 ppm in N1, 394 ppm in W1 and 397 ppm in W2 representing a particularly prominent enhancement.'

Fig 8 shows the difference between the morning and afternoon for $CO_2$ averaged across all flights. It is greater near the surface in the morning (respiration) than afternoon (photosynthesis). The role of biogenics on the $CO_2$ profiles is further discussed in P9 L18 in the updated manuscript.

P8, L31: How is the "aerosol burden" calculated? E.g. is there an inferred mass?

See comment for P8, L18.

P9, L2: Again excess aerosol should have units or some more precise explanation?

See response 7).

P9, L8-10: Here the authors do note a possible role of temperature and aging on coating thickness, which is great. Perhaps temperature and smoke age deserve broader attention?

See response 3).

P9, L11: Why the difference in coating thickness above the BL for different regimes? Do these regimes have a different concentration gradient above the BL so that evaporation due to a concentration decrease happens only in some cases?

The cause is more likely scavenging during wet convection. This point is made in response to the main recommendation of reviewer #1 – for more details see that response.

P9, L19: "influence" may be better than "determine"?

Amended

P9, L28: "combustion processes" may be better than "emissions"?

Amended

P10, L1: sentence may need one more word? E.g. "they" before "contribute"?
Amended

P10, L2-3: is the relative frequency of deep convection different from Andreae et al 2004 maybe because different geographic regions were sampled?
P10, L2-3 for reference:
"Such plumes were seldom seen above 4 km, in contrast to previous observations in the region (Andreae et al., 2004), indicating the mass flux from large pyrocumulus detrainment into the upper troposphere (within the aircraft range) was not significant."
Many of the pictorial examples given in the supplementary material of Andreae et al., 2004 seem to be within or close to our sample region in Rondonia or south of Manaus. Their location tags are slightly at odds with the flight track presented in Andreae et al., 2004 Fig. 1, but consistent with another plot of the LBA-SMOCC flight tracks in Almeida and dos Santos (2007, http://dx.doi.org/10.1590/S0102-77862007000300004). So in summary, based on visual observations only, the relative frequency of pyrocumulus was lower during SAMBBA in approximately the same sample area.
P10,L2,L3 has now been slightly amended to read:
"Such plumes were seldom seen above 4 km, in contrast to previous observations in approximately the same sample region (Supplementary material in Andreae et al., 2004), indicating the mass flux from large pyrocumulus detrainment into the upper troposphere (within the aircraft range) was not significant"

P10, L6-7: Saying pollution persisted "above" the mixed layer seems potentially inconsistent with the earlier text about clean conditions at altitude.
This section has been re-written in response to reviewer #1 – for more details see that response.

P10, L9: "they exhibit"?
Amended

P10, L13: "eats" should be "east"?
Amended

P10,L13-14: "above the lifting"?
Amended

P11, L1: Maybe "remote sensing" is too broad since MOPITT a-priori profiles, and perhaps others, are flat over the whole column?
None of the studies referred to in this section are from MOPITT retrievals.

P11. L8: replace "This study" with explicit reference to a paper?
Amended

P11, L14-28: Good points.

P12, L29-31: Christian et al 2003 (JGR), Akagi et al., (2011), Andreae & Merlet 2001, and many other papers do already recommend higher BC/OC, BC/CO, etc for savannah vs forest.
This is correct and the Hodgson et al., results are consistent with these findings. The main thrust of this point was to recommend assessment of how the significant east/west difference in pollutants propagates to climate impacts via targeted modelling studies. As such the sentence has been amended:
'Cerrado burns in the east are more flaming, whilst those in the west, of primary/secondary forest and pasture land are more smouldering (Akagi et al., 2011; Andreae and Merlet, 2001; Hodgson et al., 2018). The regional contrast in biomass burning emissions resulting from these different fire types clearly merits future investigation in modelling studies to assess potential impacts on aerosol optical properties and their radiative effects.'

P13, L10: Is this obvious from satellites or the average meteorology?
The prevailing low level easterlies are clearly identified in ECMWF reanalysis.

**Figure comments**

Figure 1. Its hard to tell the difference between high altitude surveys and straight level runs, make one orange or yellow or green?
We thank the reviewer for this suggestion. As Orange would clash with the box for 'East' and green with the LCSS land cover map, we have instead used a brown colour to ensure all features are distinguishable.

Fig 2. Should there be typical altitude values (and units) on the vertical axis?
We thank the reviewer for this suggestion. However we are of the opinion that additional altitude markers on the vertical axis will act to complicate the schematic. This is primarily because the vertical axis is not linear, but also because more precise altitudes from the layers we have observations for are provided in the accompanying legend.

Fig 3. There are two BB plumes identified on the rBC trace. There are no BB plumes identified on the CO trace, and only one each on CO2 and Bscat. It seems like there should be two on all traces if they really are BB. Also, I find it hard to locate/digest the structure and plume features that are explained in the upper right boxes.

We thank the reviewer for this comment as it illustrates the figure explanation was not adequate. A biomass burning plume was identified if any of the species passed the identified threshold (dot-dashed line). The red trace illustrates this for each species, so for example at ~650 hPa the algorithm detects a plume for rBC and the scattering coefficient, but not for CO or $CO_2$. The intention was to explicitly illustrate this by having the species with an identified plume in red whilst those without remaining in black. As one or more of the species exceeded the threshold, our scheme defines this example as a biomass burning plume. It is worth noting here that if CO is well correlated (Pearson's r > 0.5) – it is in this particular example – then the plume will be included in the scatter plot in Fig 9. C-D.

The figure label has been amended to include:
Note, the red outline of a biomass burning plume is only present if it was identified for that specific species. For example at 65 kPa rBC and $\sigma_{sp\_dry}$ pass the identification threshold (grey dot-dash line) but CO and CO2 do not.

Fig 4. How can the difference for soil moisture be in tens when the absolute values are fractional?
We thank the reviewer for pointing out this error. The values were scaled by a factor of 1000 to improve the figure appearance, but no explanation of this was provided. The text on the colour bar has now been altered to: "P1- P2 (x 1e3)"

Fig 5. Maybe label FC as "FC/100" also in Fig 6?
We thank the reviewer for this suggestion and have updated the figures accordingly.
Interesting that the AOD max is downwind of the detected fire max. This could be transport, but also different detection efficiency, higher PM emissions from more smoldering in forest fires.
We agree with the reviewer and that it is likely a mix of the suggested factors. We refrain from a detailed discussion of this in the text to ensure the manuscript remains focused on the vertical distribution.
AODs up to 4 that are mentioned in the text relate to this figure how?
The AOD of 4 referred to was an example of the extreme loadings possible as part of the introductory contextualisation. This sentence has been slightly amended for clarity from:
"Aerosol accumulates within the convective boundary layer forming a regional haze with a lifetime of 4-5 days (Edwards et al.,2006) that can cover up to 6 million km2 (Prins et al., 1998) and reach aerosol optical depths (AOD) of up to 4 in the midvisible during particularly polluted years (Artaxo et al., 2013)."
To:
"Aerosol accumulates within the convective boundary layer forming a regional haze that can cover up to 6 million km$^2$ (Prins et al., 1998) and whilst weekly averaged aerosol optical depths (AOD) are typically 0.75-1 in the mid visible, they can reach 4 in extremely polluted years associated with drought (Artaxo et al., 2013). "

Fig 6. Would dates or definitions of dry and dry-wet transition seasons be useful? Yokelson et al., (2007) saw a huge burst in burning during a few days with the lowest RH.
We thank the reviewer for this suggestion and have updated the figure captions to include the dates. Whilst an analysis of fire activity with factors such as RH would be interesting, it is outside the scope of this paper and therefore we refrain from including it.

We thank the reviewer for pointing out this error. The caption should indeed read 'meteorological' and has thus been amended.

rBC units altered to be consistent with rest of manuscript. Changed from µg sm$^{-3}$ to µg m$^{-3}$ ('s' had previously stood for 'standard').

We thank the reviewer for the suggestion and it has led us to more explicitly discount CO from urban sources as significant as the emissions hotspots are a significant distance away from our flight region and emissions are an approximately an order of magnitude lower than for biomass burning (see figure below, which has been added to the supplement). There was no influence on the average profiles from elevated CO at aircraft take-off/landing as these sections of individual profiles were removed. This is discussed in Sect. 4.1 of the manuscript.

[Figure]

Fig S9. CO emissions on 20th September 2012 from biomass burning (left) and anthropogenic sources (right). Fire CO emissions are derived from the Brazilian Biomass Burning Emission Model (3BEM). Anthropogenic CO emissions are derived from the Emissions Database for Global Atmosphere Research (EDGAR) version 4.0 2005. Both emissions maps were generated using PREP-CHEM-SRC v1.4 as described in (Archer-Nicholls et al., (2015). The dashed grey box represents the flight area and near-surface air mass history. The sum of the emissions flux within this area is 1.11 Mmol hr$^{-1}$ for fire CO and 0.29 Mmol hr$^{-1}$ for anthropogenic CO.

We thank the reviewer for bringing the lack of units to our attention. These have now been added to the figure. Unfortunately we do not understand what the "More?" comment refers to and seek clarification if the reviewer feels it important.

Coating thickness is related to deep convection and wet scavenging (Sect 4.1) and briefly, partitioning (Sect 4.1). Regarding aging, please refer to the comment 2 on page 1.

We thank the reviewer for the comment and agree it is a busy figure. We have made minor alterations to the plot in an attempt to make it clearer. In addition we have re-written the figure caption:

Figure 11. Summary of the west to east regional gradient in equivalent potential temperature (i), relative humidity (ii), horizontal wind speed (iii) and aerosol extinction (iv), based on an east to west transect (Panels B) on the 27th September 2012 (flight b743). The thermodynamic curtains (B.i-iii) are derived from linearly interpolated aircraft (grey

line) and dropsonde (grey dots) profiles. The curtain of the lidar extinction coefficient at 532 nm (B.iv) is reproduced from Marenco et al., (2016) scaled from 355nm using a scaling factor of 0.57 following Marenco et al., (2014). The regional gradient is emphasised by the side panels which show the median thermodynamic and aerosol extinction profiles for all western (A) and eastern flights (C). In panels (i) to (ii) the observed thermodynamic parameters are compared to ECMWF reanalysis data extracted along the flight paths. In panel (iv) the in-situ aerosol scattering coefficient at 550 nm is compared to the lidar extinction coefficient averaged over a similar region (Fig. S6; n.b. both measurements are reported at ambient temperature and pressure).

Some minor alterations have been made to the paragraph discussing the figure in the text:
A transect flight from east to west (Fig. 11) captures and summarises the meteorological drivers of the regional contrast in pollutant vertical distribution. A declining mixing layer depth from east to west is evident from the reduction in altitude of the sharp gradient (i.e. the entrainment zone) of $\theta_e$ from ~3 km in the east to ~1.5 km in the west. Above the mixed layer top relative humidity increases, especially so above the lifting condensation level where the high humidity distinguishes the cloud convective layer. This layer is deeper in the west, associated with deeper moist convection. A wind speed maximum is present at 5-6 km, coincident with the entrainment zone. Together, this structure can explain the lidar derived extinction coefficient distribution (first published by Marenco et al., 2016). Aerosol is capped below the first wind speed jet, well mixed within the mixed layer and features a maximum at ~1.5-2 km. Visible plumes at ~59 °W and ~52.5 °W lie at injection heights typical of those observed in the in situ profile data. The similarity of the in situ $\sigma_{sp\_amb}$ and lidar extinction coefficient profile shapes at the regional (Fig. 11) and local (Fig. S7) scales engenders confidence in the representativity of both datasets. Disparity in the absolute magnitudes is primarily explainable by differences in the sampling coverage (Fig. S8).

---

## Editor Decision (ED1)

Editor Report for "The vertical distribution of biomass burning pollution over tropical South America from aircraft in situ measurements during SAMBBA" by Eoghan Darbyshire, William T. Morgan, James D. Allan, Dantong Liu, Michael J. Flynn, James R. Dorsey, Sebastian J. O'Shea, Douglas Lowe, Kate Szpek, Franco Marenco, Ben T. Johnson, Stephane Bauguitte, Jim M. Haywood, Joel F. Brito, Paulo Artaxo, Karla M. Longo, and Hugh Coe

I have reviewed the authors' responses to the reviewers' comments and find the responses and changes in the revised manuscript satisfactory overall. Going over the revised manuscript, I found a few additional issues that will require a minor revision.

1) P. 10, L 21ff: In this paragraph, the authors report some findings and interpretation that are consistent with previous work, which should be cited here (Andreae et al., 2018): a) the trapping of pollutants by the anticyclonic circulation (Fig. 4); b) the efficient removal of aerosol particles during deep convection over Amazonia (Figs. 11b and 23); c) the vertical distribution of rBC (Figs. 15 and 18); and the transport of CO to the UT by deep convection while aerosol is depleted (Fig. 23).
2) P. 11, L 7ff: The Angstrom exponent values found in SAMBBA agree with (and sometimes are higher than) the typical dry season values reported from long-term studies in Amazonia (Rizzo et al., 2013; Saturno et al., 2018), which should be referenced here.
3) There is an extended discussion both in the paper and in the responses to the reviewers about the reasons for the increase of the aerosol/CO ratios with altitude, which shows up most prominently in the eastern region. I think the authors need to consider sources outside of the basin. It has been known for quite some time that the air that enters the basin with the SE trade winds during the dry season is by no means pristine, but contains substantial amounts of biomass emissions from southern Africa (Andreae et al., 1994; Saturno et al., 2018, and references therein). During the ACRIDICON-CHUVA 2014 aircraft campaign, we found up to 2 $\mu$g m$^{-3}$ of rBC in layers at altitudes between 3 and 4 km. The savanna fires in Africa produce among the highest BC/CO ratios found anywhere (flaming dominates), and the layers observed during AC had a $\Delta$rBC/$\Delta$CO of about 20. Mixing this material with the smoke generated in the Amazon can easily generate the upward positive gradient found by Darbyshire et al. This potential explanation should be addressed in their paper.

Andreae, M. O., Anderson, B. E., Blake, D. R., Bradshaw, J. D., Collins, J. E., Gregory, G. L., Sachse, G. W., and Shipham, M. C.: Influence of plumes from biomass burning on atmospheric chemistry over the equatorial Atlantic during CITE-3, J. Geophys. Res., 99, 12,793-12,808, 1994.
Andreae, M. O., Afchine, A., Albrecht, R., Holanda, B. A., Artaxo, P., Barbosa, H. M. J., Borrmann, S., Cecchini, M. A., Costa, A., Dollner, M., Fütterer, D., Järvinen, E., Jurkat, T., Klimach, T., Konemann, T., Knote, C., Krämer, M., Krisna, T., Machado, L. A. T., Mertes, S., Minikin, A., Pöhlker, C., Pöhlker, M. L., Pöschl, U., Rosenfeld, D., Sauer, D.,

Schlager, H., Schnaiter, M., Schneider, J., Schulz, C., Spanu, A., Sperling, V. B., Voigt, C., Walser, A., Wang, J., Weinzierl, B., Wendisch, M., and Ziereis, H.: Aerosol characteristics and particle production in the upper troposphere over the Amazon Basin, Atmos. Chem. Phys., 18, 921-961, doi:10.5194/acp-18-921-2018, 2018.

Rizzo, L. V., Artaxo, P., Müller, T., Wiedensohler, A., Paixão, M., Cirino, G. G., Arana, A., Swietlicki, E., Roldin, P., Fors, E. O., Wiedemann, K. T., Leal, L. S. M., and Kulmala, M.: Long term measurements of aerosol optical properties at a primary forest site in Amazonia, Atmos. Chem. Phys., 13, 2391-2413, doi:10.5194/acp-13-2391-2013, 2013.

Saturno, J., Holanda, B. A., Pöhlker, C., Ditas, F., Wang, Q., Moran-Zuloaga, D., Brito, J., Carbone, S., Cheng, Y., Chi, X., Ditas, J., Hoffmann, T., Hrabe de Angelis, I., Könemann, T., Lavrič, J. V., Ma, N., Ming, J., Paulsen, H., Pöhlker, M. L., Rizzo, L. V., Schlag, P., Su, H., Walter, D., Wolff, S., Zhang, Y., Artaxo, P., Pöschl, U., and Andreae, M. O.: Black and brown carbon over central Amazonia: long-term aerosol measurements at the ATTO site, Atmos. Chem. Phys., 18, 12817-12843, doi:10.5194/acp-18-12817-2018, 2018.

---

## Author Response (AR2)

"The vertical distribution of biomass burning pollution over tropical South America from aircraft in situ measurements during SAMBBA" by Eoghan Darbyshire, William T. Morgan, James D. Allan, Dantong Liu, Michael J. Flynn, James R. Dorsey, Sebastian J. O'Shea, Douglas Lowe, Kate Szpek, Franco Marenco, Ben T. Johnson, Stephane Bauguitte, Jim M. Haywood, Joel F. Brito, Paulo Artaxo, Karla M. Longo, and Hugh Coe

**Author's Response to Editor Report**

We thank the Editor for his thorough evaluation and constructive recommendations which we feel have improved this manuscript.

1) P. 10, L 21ff: In this paragraph, the authors report some findings and interpretation that are consistent with previous work, which should be cited here (Andreae et al., 2018): a) the trapping of pollutants by the anticyclonic circulation (Fig. 4); b) the efficient removal of aerosol particles during deep convection over Amazonia (Figs. 11b and 23); c) the vertical distribution of rBC (Figs. 15 and 18); and the transport of CO to the UT by deep convection while aerosol is depleted (Fig. 23).

We thank the editor for the reference which covers a very interesting and relevant study we were not aware of. The relevant sections have been updated to include reference to this study (see tracked manuscript).

2) P. 11, L 7ff: The Angstrom exponent values found in SAMBBA agree with (and sometimes are higher than) the typical dry season values reported from long-term studies in Amazonia (Rizzo et al., 2013; Saturno et al., 2018), which should be referenced here.

We thank the editor for these references and have now included them when discussing the Angstrom exponent (see tracked manuscript). We have not gone into detailed comparison to maintain the focus of the paper, especially given section 4.1 is already so lengthy.

3) There is an extended discussion both in the paper and in the responses to the reviewers about the reasons for the increase of the aerosol/CO ratios with altitude, which shows up most prominently in the eastern region. I think the authors need to consider sources outside of the basin. It has been known for quite some time that the air that enters the basin with the SE trade winds during the dry season is by no means pristine, but contains substantial amounts of biomass emissions from southern Africa (Andreae et al., 1994; Saturno et al., 2018, and references therein). During the ACRIDICON-CHUVA 2014 aircraft campaign, we found up to 2 µg m-3 of rBC in layers at altitudes between 3 and 4 km. The savanna fires in Africa produce among the highest BC/CO ratios found anywhere (flaming dominates), and the layers observed during AC had a ΔrBC/ΔCO of about 20. Mixing this material with the smoke generated in the Amazon can easily generate the upward positive gradient found by Darbyshire et al. This potential explanation should be addressed in their paper.

We thank the editor for this insight. A brief look at CALIPSO profiles many years ago led us to believe that importation into the basin was not significant, however upon revisiting these lidar curtains we can now see this thought was misguided. Specifically, looking at the curtains offshore and upwind of E0 shows that, at times, there are layers of smoke which have significant extinction coefficients of around 50 Mm$^{-1}$ (based on a quick, back of the envelope conversion). These layers are present all the way to the active burning regions of Western and Southern Africa, but the trajectories indicate the likely source is from Western Africa. These CALIPSO lidar curtains are now included in the supplement, Fig. S13. Coupled to the Editors insight from the ACRIDICON-CHUVA 2014 flights and the unpublished biomass burning observations in Africa from the Manchester group (CLARIFY and MOYA projects) we can see that BC rich material is transported to the Amazon – a detailed investigation into the frequency and thus significance of this phenomenon will represent an interesting future study. The abstract/conclusions have been amended accordingly, and the following text has been added to the discussion section:

[revised manuscript text omitted]